# Mitigating the Backdoor Effect for Multi-Task Model Merging via Safety-Aware Subspace

**Jinluan Yang**[1]  **Anke Tang**[2]  **Didi Zhu**[1]  **Zhengyu Chen**[1]  **Li Shen**[3*]  **Fei Wu**[1*]
[1]Zhejiang University  [2]Whuhan University  [3]Shenzhen Campus of Sun Yat-sen University
{yangjinluan, didi_zhu, chenzhengyu, wufei}@zju.edu.cn
anketang@whu.edu.cn, mathshenli@gmail.com

## Abstract

Model merging has gained significant attention as a cost-effective approach to integrate multiple single-task fine-tuned models into a unified one that can perform well on multiple tasks. However, existing model merging techniques primarily focus on resolving conflicts between task-specific models, they often overlook potential security threats, particularly the risk of backdoor attacks in the open-source model ecosystem. In this paper, we first investigate the vulnerabilities of existing model merging methods to backdoor attacks, identifying two critical challenges: backdoor succession and backdoor transfer. To address these issues, we propose a novel *Defense-Aware Merging (DAM)* approach that simultaneously mitigates task interference and backdoor vulnerabilities. Specifically, DAM employs a meta-learning-based optimization method with dual masks to identify a shared and safety-aware subspace for model merging. These masks are alternately optimized: the Task-Shared mask identifies common beneficial parameters across tasks, aiming to preserve task-specific knowledge while reducing interference, while the Backdoor-Detection mask isolates potentially harmful parameters to neutralize security threats. This dual-mask design allows us to carefully balance the preservation of useful knowledge and the removal of potential vulnerabilities. Compared to existing merging methods, DAM achieves a more favorable balance between performance and security, reducing the attack success rate by 2-10 percentage points while sacrificing only about 1% in accuracy. Furthermore, DAM exhibits robust performance and broad applicability across various types of backdoor attacks and the number of compromised models involved in the merging process. Our codes and models can be accessed through DAM.

## 1 Introduction

The rapid advancement of artificial intelligence has led to the emergence of pre-trained models that demonstrate exceptional performance across various tasks (Yang et al., 2024a). However, training and deploying individual models for each specific task not only incurs substantial computational costs but also results in knowledge redundancy and storage inefficiencies. To address these challenges, multi-task model merging, as a promising solution, integrates parameters from multiple single-task models into a unified model (Tang et al., 2024a), which not only enhances task-specific performance but also significantly improves computational efficiency and cost-effectiveness (Izmailov et al., 2018; Frankle et al., 2020; Ilharco et al., 2022b).

Current research in model merging primarily focuses on resolving conflicts among task-specific models to achieve effective knowledge transfer and inheritance. Pioneering merging strategies based on task vectors

---

*Corresponding Authors.

include gradient conflict-based methods (Yadav et al., 2024) and subspace-based approaches (Tang et al., 2023; Tam et al., 2024; Yu et al., 2024). However, in the pursuit of performance optimization, these methods often neglect critical security considerations, particularly the risk of backdoor attacks. The open-source model ecosystem (Liu et al., 2024) facilitates frequent model downloading, fine-tuning, and re-uploading by users. While this practice enhances knowledge dissemination and collaborative development, it simultaneously introduces potential security vulnerabilities. Malicious actors may exploit this ecosystem by uploading models injected with backdoors. When these compromised models are incorporated into multi-task merging processes, the resultant merged model may produce misguided outputs in response to specific trigger inputs. This scenario poses a significant threat to the integrity and reliability of the entire open-source ecosystem. Consequently, we urgently need to address two critical questions in the model merging process:

*Have existing multi-task merging methods adequately addressed these overlooked security issues?*

*If not, how can we better mitigate the backdoor effects in multi-task merging?*

In this paper, we first revisit current multi-task merging strategies and evaluate their performance and safety when merging potentially compromised single-task models (See more analysis details in Section 2.3). This analysis reveals two critical issues previously overlooked: *backdoor succession* and *backdoor transfer*. *Backdoor succession* refers to the phenomenon where the harmful elements from one or more backdoored models still persist in the merged model, while *backdoor transfer* describes the propagation of these harmful elements from backdoored models to clean models, affecting the security and performance of clean models during the merging process. Both of them pose the security risk of the multi-task merging process, highlighting the necessity for a safety-aware model merging approach that not only maximizes performance but also ensures the safety of the merged model.

To address these challenges, we propose a Defense-Aware Merging (DAM) algorithm that simultaneously mitigates task interference and backdoor issues by identifying a shared and safe-aware subspace. To achieve this dual objective, we develop a meta-learning-based optimization method employing two specialized masks: a Task-Shared Mask and a Backdoor-Detection Mask. The Task-Shared Mask identifies shared parameter subspaces across different tasks, aiming to preserve task-specific knowledge while mitigating interference between tasks. Concurrently, the Backdoor Detection Mask is designed to detect parameters potentially associated with backdoor threats, isolating and neutralizing harmful elements that might have been introduced through contaminated models. These two masks are optimized alternately in an iterative process. After the alternating optimization, we reset the parameters within the full mask region of task vectors to their pre-trained weights to develop a merged model that effectively balances performance and safety.

Through extensive experiments, DAM demonstrates superior performance by reducing attack success rates by 2-10% across various scenarios, while sacrificing only about 1% in accuracy, thus achieving a more favorable balance between performance and security. Furthermore, DAM exhibits robust performance and broad applicability across various types of backdoor attacks and the number of backdoored models involved in the merging process. In summary, the contributions of this paper are threefold:

- We reveal for the first time the vulnerabilities of current multi-task merging methods under backdoor attacks, identifying "backdoor succession" and "backdoor transfer" as core challenges.

- We propose a novel Defense-Aware Merging (DAM) algorithm through a dual-mask optimization, that not only mitigates task interference but also effectively alleviates the backdoor effect for the multi-task model merging process.

- Extensive experiments validate the effectiveness of DAM, demonstrating significant performance improvements across multiple benchmarks and backbone networks while maintaining minimal accuracy degradation.

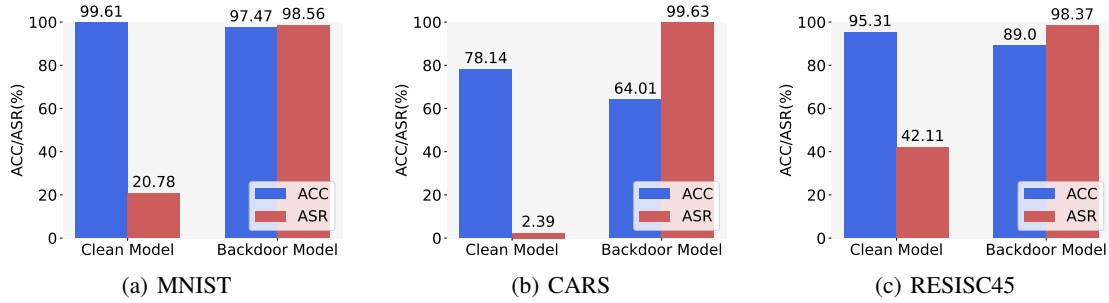

Figure 1: Performance comparison between clean and backdoor(TrojVit) adopting CLIP-ViT-B/32.

## 2 EXPLORING THE BACKDOOR EFFECT DURING MODEL MERGING

In this section, we first provide an overview of existing multi-task merging techniques and discuss the difference between optimized objects while merging considering the existence of backdoor. Then, extensive experiments are conducted to unpack the phenomenon of backdoor succession and backdoor transfer.

### 2.1 PRELIMINARIES

**Previous Multi-Task Merging Techniques.** Denote the $f_\theta$ as the CLIP-like pre-trained model $f$ with weights $\theta$ and a set of datasets $\mathcal{D} = \{D_i\}_{i=1}^n$ for $n$ downstream tasks. We can fine-tune the pre-trained model parameterized by $\theta_{\text{pre}}$ to acquire $n$ task-specific models parameterized by $\{\theta_i\}_{i=1}^n$. Then, for each task $i$, the task vector can defined as the difference between $\theta_{\text{pre}}$ and $\theta_i$, i.e., $\tau_i = \theta_i - \theta_{\text{pre}}$. Existing merging methods can be formulated as the optimization to acquire $\theta_{merged} = \theta_{\text{pre}} + \sum_{i=1}^n \{\lambda_i \tau_i'\}$, where $\forall \lambda \in [0, 1]$ refers to the merging coefficient and $\phi(\tau_i) = \tau_i'$ represents the revision for each task vector. The main difference among these methods exists in ways to acquire the $\tau_i'$ and $\lambda_i$. For example, both Weight Average (Wortsman et al., 2022) and Task Arithmetic (Ilharco et al., 2022a) adopt the origin task vector $\tau_i$, with the $\lambda = \frac{1}{n}$ adapted to the number of tasks and a fixed $\lambda = 0.3$ respectively. Ties-Merging (Yadav et al., 2024) and Concrete (Tang et al., 2023) address the interference among tasks and replace the original task vector with $\tau_i'$. Moreover, RegMean (Jin et al., 2022) and AdaMerging (Yang et al., 2023) respectively formulate the optimization of $\lambda_i$ according to the model's activations and the entropy on an unlabeled held-out dataset. However, these works share the same and single optimization objective to maximize the performance of the merged model on the clean test datasets as Eq.1 from the evaluation perspective, where $\mathcal{A}$ is a model merging algorithm associated with $\phi(\cdot)$ and $\lambda$. It is uncertain if current merging methods remain effective considering safety issues like backdoors, which introduces potential but important concerns for deploying merging algorithms to more scenarios.

$$\max_{\phi, \lambda} \frac{1}{n} \sum_{i=1}^n \text{Performance} \left( \mathcal{A}(\theta_{\text{pre}}, \phi(\tau_i), \lambda_i), D_i^{\text{test}} \right). \tag{1}$$

**Merging Considering the Existence of Backdoor.** Typically, a model injected with the backdoor behaves normally for clean input data but will be misguided toward the target class when the inputs contain a specific trigger (Wu et al., 2022). Thus, when backdoored task-specific models are utilized during merging, the Eq.1 should also be rewritten as Eq.2. Universally, the accuracy (ACC) represents the percentage of test input images without triggers classified into their corresponding correct classes (true label), while the attack success rate (ASR) shows the percentage of input images embedded with a trigger classified into the pre-defined target class (label decided by the attackers) (Wu et al., 2022). Ideally, the optimization of model merging

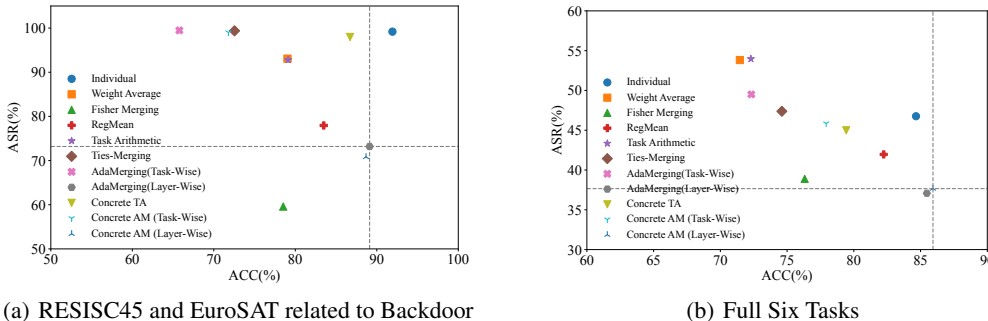

(a) RESISC45 and EuroSAT related to Backdoor  (b) Full Six Tasks

Figure 2: Backdoor Succession Evaluation: Average performance on multi-tasks while merging two back-doored task-specific models (RESISC45 and EuroSAT) and four clean task-specific models (MNIST, CARS, SVHN and DTD). The grey line shows the SOTA multi-task merging technique, but its ASR still exceeds 70% on tasks related to the backdoor and 35% on full tasks though achieves great performance(ACC).

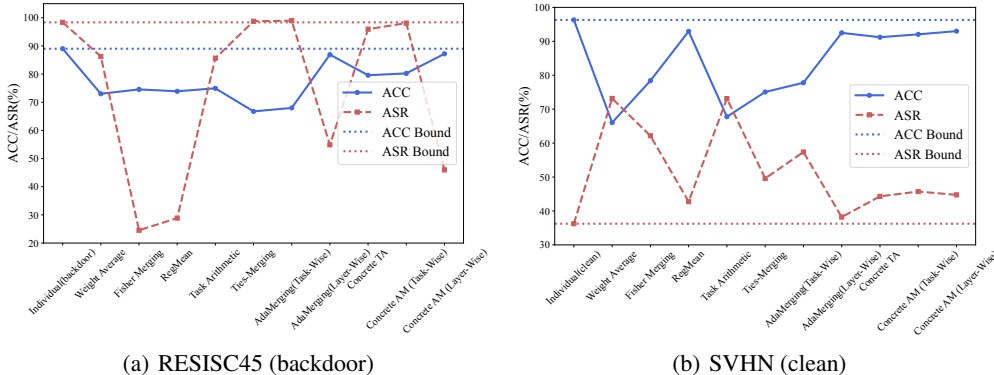

(a) RESISC45 (backdoor)  (b) SVHN (clean)

Figure 3: Backdoor Transfer Evaluation: Single-task performance while merging two backdoored task-specific models (RESISC45 and EuroSAT) and four clean task-specific models (MNIST, CARS, SVHN and DTD). The ACC Bound and ASR Bound can be set according to the clean or backdoored individual fine-tuned models. The ideal merged model should be close or even upper to the ACC Bound and lower or at least close to the ASR Bound, but different merging methods exhibit unexpected trends due to the backdoor transfer.

should be towards high ACC and low ASR simultaneously to maximize safety and performance considering the existence of the backdoor. The $\omega$ is set as the balance weight for the performance and safety by default.

$$\max_{\phi,\lambda} \frac{1}{n} \sum_{i=1}^{n} \left( \text{Performance} \left( \mathcal{A}(\theta_{\text{pre}}, \phi(\tau_i), \lambda_i), D_i^{\text{test}} \right) + \omega \cdot \text{Safety} \left( \mathcal{A}(\theta_{\text{pre}}, \phi(\tau_i), \lambda_i), D_i^{\text{test with trigger}} \right) \right) \quad (2)$$

## 2.2 THE SETTINGS OF MODEL MERGING CONSIDERING THE BACKDOOR

Based on Eq.2, we further explore whether existing merging methods can naturally cope with neglected backdoor issues. Specifically, we take the CLIP-ViT (Radford et al., 2021) as the pre-trained model and explore the backdoor effect during model merging on image classification tasks (Tang et al., 2024a).

**Backdoored Model Constructions:** We utilize the commonly used vit-specific (patch-wise) backdoor attack, TrojVit (Zheng et al., 2023), to construct backdoored models. As shown in Figure 1 and Figure 6, backdoored

models achieve good ACC but higher (worse) ASR than clean ones, bringing the potential risk to model merging. The detailed construction implementations can be shown in Appendix C.1.

**Attack and Defense Scenario:** We assume that the adversary is the provider of backdoored models and doesn't know other task-specific models and merging algorithms. For defenders, they only have different checkpoints without the knowledge of whether or not and which region they are injected with a backdoor.

### 2.3 Unpacking the Phenomenon of Backdoor Succession and Backdoor Transfer

The original object of multi-task merging is to provide a cost-effective parameter-level fusion strategy to obtain a multi-task model that can achieve close or better performance than individual fine-tuned models for each task (Yang et al., 2024b). While considering the existence of the backdoor during merging, this object can be transformed as *Promote the merged model close or even upper to the ACC of clean individual fine-tuned models and lower or at least equal to the ASR of the backdoored individual fine-tuned models.* Thus, we explore the backdoor effect in multi-task merging by comparing the ACC and ASR of the merged model with those of individual fine-tuned models across tasks. We have the following two important findings.

**Backdoor Succession During Merging:** From the Figure 2 and Figure 7, we can observe that the development of merging techniques has increased the ACC of merged model close or even better than individual fine-tuned models. While considering the safety issues, the ASR of the merged model on the backdoor-related task (e.g. EUROSAT, shown in Table 14 ) decreases but is still high, due to the backdoor succession as *Finding 1*.

> **Finding 1**: Backdoor Succession: The backdoor effect from backdoored task-specific models can not be mitigated well though we have adopted existing state-of-the-art techniques during multi-task merging.

**Backdoor Transfer During Merging:** The Figure 3 and Figure 8 describe that though we provide clean task-specific models for merging (e.g. SVHN shown in Table 13), the ASR of the merged model on SVHN will unexpectedly increase compared with the individual fine-tuned model on SVHN, due to other backdoored task-specific models. This can be accounted for by *Finding 2*. Specifically, from the perspective of parameter disentanglement, different task-specific models have a common parameter region, if this task-general region is injected with the backdoor, the backdoor effect can be seen as transferring from backdoored models to clean models during model merging. More detailed discussions can be shown in the Appendix B.5.

> **Finding 2**: Backdoor Transfer: The backdoor effect from backdoored task-specific models can transfer to other clean task-specific models, which leads to special safety-related challenges for multi-task merging.

## 3 Defense-Aware Merging

Based on the analysis shown in Section 2, we face two challenges to achieve a merged model with high ACC and low ASR: (i) How can we cope with backdoor issues during merging when we can not know the backdoor types and whether the models that need merging are safe or not in advance? (ii) How can we unify the optimization process to achieve a good trade-off between performance and safety during merging?

**To address (i)**, we respectively synthesize a universal perturbation for each task-specific model to represent undesired behavioral changes from trigger insertion, without requiring assumptions about backdoor information (e.g. the trigger's size or location) (Zeng et al., 2024). Assisted by the synthesized perturbations, we can identify and adjust the parameters related to the backdoor during merging, assuming that the backdoor-related parameters are more sensitive to the perturbations (Wu et al., 2022). **To address (ii)**, we provide a dual-mask optimization strategy to identify a shared and safety-aware subspace on task vectors, which represents the low-dimensional-parameter area compared with the full parameters of task vectors, aiming to concurrently mitigate interference and backdoor issues during merging with the safe and unsafe components discerned using the learned perturbations from (i), achieving a good trade-off between performance and safety.

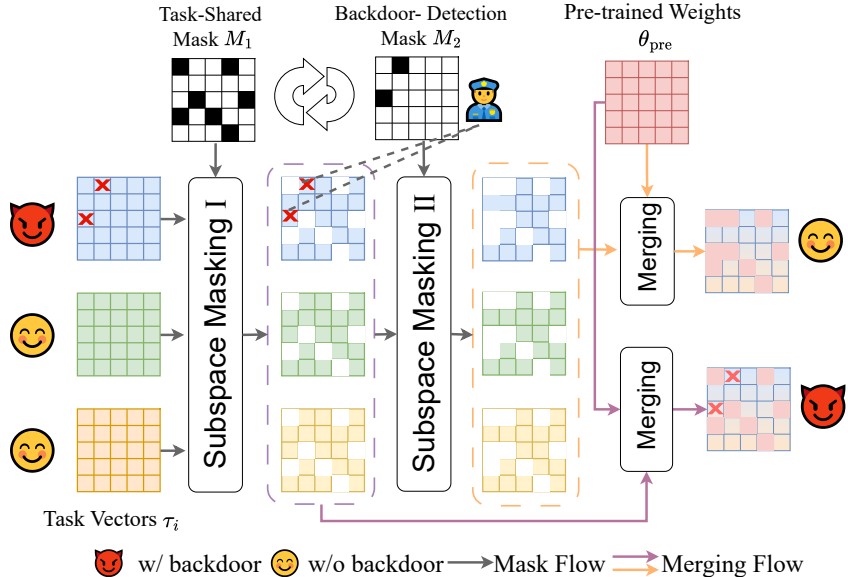

Figure 4: Illustrations of Defense-Aware Merging(DAM), where the Task-Shared mask and Backdoor-Detection mask are respectively used to mitigate the interference issues existing in the task-shared parameters among models and the safety issues existing in the task-specific parameters from the backdoored models.

Thus, the framework of DAM can be formulated as a bi-level optimization problem as Eq. 3, where $\mathcal{A}$ is the merging operation associated with $\lambda$ and two different mask designs (Task-Shared Mask $M_1$ and Backdoor-Detection Mask $M_2$), where $M_1, M_2 \in \mathbb{R}^d$ referring to the $\phi(\cdot)$ in Eq 1 to revise the task vectors $\tau$. The $M_1 \odot M_2$ aims to achieve a union set for two masks through the element multiplication. The $\alpha$ is a balance weight for mask optimization, with larger $\alpha$ values favoring safety over performance.

$$\min_{M_1, M_2} \sum_{i=1}^{n} \left[ \mathcal{L}\left(\mathcal{A}(\theta_{\text{pre}}, \{M_1 \odot \tau_j\}_{j=1}^{n}, \lambda^*), D_i\right) + \alpha \mathcal{L}\left(\mathcal{A}(\theta_{\text{pre}}, \{M_1 \odot M_2 \odot \tau_j\}_{j=1}^{n}, \lambda^*), D_i + \Delta_i^*\right) \right] \quad (3)$$

$$\text{s.t.} \begin{cases} \lambda^*(M_1) = \arg\min_{\lambda} \sum_{i=1}^{n} \mathcal{L}\left(\mathcal{A}(\theta_{\text{pre}}, \{M_1 \odot \tau_j\}_{j=1}^{n}, \lambda), D_i\right), \\ \Delta_i^*(M_1, M_2) = \arg\min_{\Delta_i} \mathcal{L}\left(\mathcal{A}(\theta_{\text{pre}}, \{M_1 \odot M_2 \odot \tau_j\}_{j=1}^{n}, \lambda^*), D_i + \Delta_i\right), \end{cases}$$

The $\mathcal{L}$ is usually defined as the unsupervised loss such as entropy loss as Eq. 4 for test-time adaptation under the share-and-play scenario (Yang et al., 2023) where we have no access to the training data. For image classification tasks, we can initially employ the merged model to generate predictions $\hat{y}$ on the unlabeled test data and subsequently utilize these predictions to optimize the merged model, where $x_i$ is the $i$-th unlabeled sample, $p(\hat{y}_c|x_i)$ is the predicted probability of the $c$-th class, and $C$ is the number of classes.

$$\mathcal{L}_{\text{entropy}} = \mathbb{E}[-\log p(\hat{y}|x)] = -\frac{1}{n} \sum_{i=1}^{n} \sum_{j=1}^{C} p(\hat{y}_c|x_i) \log p(\hat{y}_c|x), \quad (4)$$

**The Outer-Level Optimization** aims to find a shared and safety-aware subspace across different task vectors $\{\tau_j\}_{j=1}^{n}$ that minimizes the loss of the merged model across clean and perturbed test data. Specifically, this

can be achieved through a dual-mask optimization strategy. The first term refers to the update of $M_1$ on the clean test data, aiming to improve merged model performance on multi-tasks. In contrast, the second term shows that, based on $M_1$, $M_2$ is additionally designed to lower the weight related to identified triggers to mitigate the backdoor, assisted by the synthesized perturbation in the inner-level optimization.

We can verify the reasonability of this dual-mask strategy from two aspects: (i) From the perspective of *Model Merging*, the design of $M_1$ assumes that the interference among tasks is the key to influencing merged model performance and it usually exists in the shared parameter space among different individual finetuned models (Tang et al., 2023). As shown in Figure 4: The $M_1$ is designed to identify the shared parameter space among different task vectors. Then, the task-specific parameters can be separated after Subspace Masking I. Following the purple Merging Flow, the parameter of the final merged model includes two parts: the merged version of these task-specific parameters and the parameter from the pre-trained model corresponding to the shared mask $M_1$ region. However, though it has been verified that revising the task vectors through mask designs similar to $M_1$ can improve the merged model performance clearly (Tang et al., 2024a), the backdoor will still succeed as shown in the outcome of the purple Merging Flow. That's because the backdoor-related parameters (the red cross) may exist in the task-specific parameter space (blue part). Thus, we propose to utilize an additional mask $M_2$ to identify these backdoor-related parameters and then integrate with $M_1$ to acquire the shared and safety-aware subspace. After the Subspace Masking II, following the yellow Merging Flow, the backdoor effect from fine-tuned models can be mitigated by replacing the backdoor-related parameters with pre-trained weights. But notably, the introduced $M_2$ will also lose part of the useful task-specific parameter (the green and yellow part existing in the represented region of $M_2$), leading to a decrease in model performance. Thus, *the alternate optimization* of two masks can be seen as seeking a good trade-off between performance and safety during merging; (ii) From the perspective of *Pareto Optimal Balance between performance and safety* during the dual mask optimization, we propose Theorem 1.

**Theorem 1** (Existence of Pareto front). *Let $P(\mathbf{M})$ and $S(\mathbf{M})$ denote the performance and safety measures of the merged model under mask $\mathbf{M} = (M_1, M_2)$, respectively. There exists a Pareto front $\mathcal{F}$ such that:*

$$\mathcal{F} = \{(\mathbf{M}, P(\mathbf{M}), S(\mathbf{M})) \mid \nexists \mathbf{M}' : P(\mathbf{M}') > P(\mathbf{M}) \wedge S(\mathbf{M}') > S(\mathbf{M})\}. \tag{5}$$

The detailed proof can be shown in Appendix B.3. Moreover, we also provide the convergence analysis to discuss how DAM converges to a Pareto optimal solution to balance performance and safety.

**The Inner-Level Optimization** has two objects:(i) Find the optimal merging coefficient $\lambda$ that minimizes the loss of the merged model across different tasks on clean test data as Eq.4; (ii) Estimate the trigger pattern through synthesized adversarial perturbation $\Delta$, which can be used for learning the $M_2$ to identify the sensitive backdoor-related weight in outer-level optimization. Notably, during inner-level optimization, we first optimize the $\lambda$ to get $\lambda^*$ and then create the merged model for the second objects.

Especially, the synthesized unified perturbation can be used to identify the backdoor-related parameter without additional assumptions about the injected backdoor. That's because, as shown by Eq.6, for each task data $D_i$, the adversarial samples from a backdoored model have similar features as the triggered images, but the ones from a clean model don't have this property (Wei et al., 2023; Niu et al., 2024).

$$f_{\text{clean}}(D_i + \Delta_i) \neq f_{\text{backdoor}}(D_i + \Delta_i) \approx f_{\text{backdoor}}(D_i^{\text{with trigger}}) \tag{6}$$

At the same time, the adversarial examples produced based on the backdoored models come from arbitrary classes, usually exhibiting a uniform distribution in the embedding space. Leveraging the embedding drift insight (Zeng et al., 2024) that backdoor triggers induce relatively uniform drifts in the model's embedding space regardless of the trigger location or attack mechanism, we can synthesize a unified perturbation to represent the misguided behavior change upon trigger insertion for each task-specific model. Thus, the unknown backdoor injections can still be successfully approximated as a unified and synthesized perturbation.

# 4 EXPERIMENTS

## 4.1 EXPERIMENTAL SETUP

**Datasets and Models:** Following (Tang et al., 2024a), we utilize CLIP-ViT-B/32 and CLIP-ViT-L/14 as our pre-trained models and conduct experiments on six image classification tasks including Stanford Cars (Krause et al., 2013), RESISC45 (Cheng et al., 2017), EuroSAT (Helber et al., 2018), SVHN (Netzer et al., 2021), MNIST (Lecun et al., 1998) and DTD (Cimpoi et al., 2014). We first construct the clean individual fine-tuned models by directly fine-tuning the pre-trained model on these clean datasets and then inject them with the backdoor adopting TrojVit (Zheng et al., 2023) and BadVit (Yuan et al., 2023) strategy to construct the backdoored models. More detailed descriptions of datasets and models can be shown in the Appendix C.1.

**Baselines::(i) Individual Finetuning:** All Clean Finetuned Models, All Backdoored Finetuned Models, and Mixing with Clean and Backdoored Finetuned Models under different settings. Notably, we just average their results for reference;**(ii) Multi-Task Merging Methods:** Weight Average (Wortsman et al., 2022), Fisher merging (Matena & Raffel, 2022), RegMean (Jin et al., 2022), Task Arithmetic (Ilharco et al., 2022a), Ties-Merging (Yadav et al., 2024), Adamerging (Yang et al., 2023), Concrete (Tang et al., 2023) **(iii) Post-Defense Methods** involving adversarial perturbation: ANP (Wu & Wang, 2021), AWM (Chai & Chen, 2022) and SAU (Wei et al., 2023)). Notably, we execute these post-defense backdoor processing on the best-merged model adopting (ii), which can be seen as *two-stage methods* compared with our *end-to-end training process*. **(iv) Other backdoor-related merging works that are not designed for multi-task merging**: Both WAG (Arora et al., 2024) and LoRA-as-an-Attack (Liu et al., 2024) defend the backdoor by directly averaging the homogeneous clean and backdoored full model weights or LoRa *on the same task* without other complex designs (e.g.subspace). Moreover, the BadMerging (Zhang et al., 2024) is a newly proposed backdoor attack adapted to model merging. More detailed discussions can be shown in Appendix A.

**Evaluation Metric:** We respectively adopt the top-1 accuracy(ACC) on clean test data as a performance metric and the attack success rate(ASR) on test data with trigger as a safety metric (Wu et al., 2022). An ideal model should have high ACC but low ASR. To further explore the backdoor effect, apart from the average ACC and ASR on the full six tasks, we also present the average results on tasks related to backdoored task-specific models, including ACC(2)/ASR(2) and ACC(4)/ASR(4), with the numbers indicating the count of backdoored models.

Table 1: Necessary specifications for the implementation and properties of each method.

| METHOD | TRAINING-DATA TUNING | VALID-DATA TUNING INPUTS | VALID-DATA TUNING LABLES | SAFETY-AWARE TRAINING | POST-HOC COST |
|---|---|---|---|---|---|
| Weight Average | × | × | × | × | × |
| Fisher-Merging | × | ✓ | × | × | × |
| RegMean | × | ✓ | × | × | × |
| Task Arithmetic | × | ✓ | ✓ | × | × |
| Ties-Merging | × | ✓ | ✓ | × | × |
| AdaMerging | × | ✓ | × | × | × |
| Concrete | × | ✓ | × | × | × |
| ANP | × | ✓ | ✓ | ✓ | ✓ |
| AWM | × | ✓ | ✓ | ✓ | ✓ |
| SAU | × | ✓ | ✓ | ✓ | ✓ |
| **DAM(Ours)** | × | ✓ | × | ✓ | × |

**Multi-Task Merging Settings Considering the Existence of Backdoor.** To help understand our contribution, we provide a clear overview of existing merging methods and potential post-defense solutions that can address the backdoor issues during multi-task merging. Detailed information about the implementation and properties of methods can be shown in Table 1. In short, we provide a cost-effective and safety-aware merging method to mitigate the neglected backdoor issues during multi-task merging.

## 4.2 EXPERIMENTAL RESULTS

*DAM can outperform state-of-art multi-task merging methods to achieve a better trade-off between performance and safety.* We respectively conduct multi-task model merging experiments on CLIP-ViT-B/32 and CLIP-Vit-L/14, where exists two backdoored task-specific models and four clean models. The obtained results can be shown in Table 2 and Table 10. We can observe that DAM can achieve lower ASR with a minor sacrifice of ACC compared with the SOTA merging method, Concrete AM(layer-wise). More multi-task

Table 2: Results of multi-task merging while adopting two models attacked by TrojVit (CLIP-ViT-B/32, ACC↑/ASR↓). We highlight the best average score in bold and the second score with underline.

| METHOD | MNIST(clean) ACC | ASR | Cars(clean) ACC | ASR | RESISC45(backdoor) ACC | ASR | EuroSAT(backdoor) ACC | ASR | SVHN(clean) ACC | ASR | DTD(clean) ACC | ASR | AVG ACC(2) | ACC(6) | ASR(2) | ASR(6) |
|---|---|---|---|---|---|---|---|---|---|---|---|---|---|---|---|---|
| Individual(All Clean) | 99.60 | 20.78 | 78.14 | 2.39 | 95.30 | 42.11 | 99.07 | 24.37 | 96.30 | 36.24 | 78.72 | 22.77 | 97.19 | 91.19 | 33.24 | 24.78 |
| Individual(All Backdoor) | 97.47 | 98.56 | 64.00 | 99.63 | 89.00 | 98.37 | 94.85 | 100.00 | 84.86 | 89.67 | 61.50 | 98.90 | 91.93 | 81.95 | 99.19 | 97.52 |
| Individual(Two Backdoor) | 99.60 | 20.78 | 78.14 | 2.39 | 89.00 | 98.37 | 94.85 | 100.00 | 96.30 | 36.24 | 78.72 | 22.77 | 91.93 | 89.44 | 99.19 | 46.76 |
| Weight Average | 89.51 | 36.10 | 63.35 | 1.37 | 73.00 | 86.16 | 85.07 | 100.00 | 66.04 | 73.15 | 51.76 | 26.12 | 79.04 | 71.46 | 93.08 | 53.82 |
| Fisher Merging | 97.18 | 25.18 | 65.58 | 1.64 | 74.56 | 24.54 | 82.48 | 94.63 | 78.38 | 62.16 | 59.73 | 25.21 | 78.52 | 76.32 | 59.59 | 38.89 |
| RegMean | 98.37 | 19.45 | 69.11 | 2.42 | 73.89 | 58.86 | 93.08 | 97.04 | 92.94 | 42.74 | 66.06 | 31.22 | 83.49 | 82.24 | 77.95 | 41.96 |
| Task Arithmetic | 91.30 | 35.67 | 63.60 | 1.34 | 74.94 | 85.65 | 83.30 | 100.00 | 67.76 | 73.12 | 52.82 | 25.64 | 79.12 | 72.29 | 92.83 | 53.57 |
| Ties-Merging | 98.04 | 16.25 | 63.64 | 0.40 | 66.75 | 98.76 | 78.33 | 100.00 | 85.06 | 49.59 | 55.85 | 19.26 | 72.54 | 74.61 | 99.38 | 47.38 |
| AdaMerging(Task-Wise) | 97.83 | 18.46 | 63.80 | 1.01 | 67.98 | 98.95 | 63.56 | 100.00 | 77.79 | 57.37 | 62.98 | 21.28 | 65.77 | 72.32 | 99.48 | 49.51 |
| AdaMerging(Layer-Wise) | 98.19 | 13.51 | 73.73 | 0.97 | 86.90 | 54.87 | 91.33 | 91.51 | 92.49 | 38.21 | 70.16 | 23.40 | 89.12 | 85.47 | 73.19 | 37.08 |
| Concrete TA | 98.43 | 16.37 | 61.62 | 0.34 | 79.59 | 95.95 | 93.85 | 100.00 | 91.20 | 44.30 | 51.86 | 12.98 | 86.72 | 79.43 | 97.98 | 44.99 |
| Concrete AM (Task-Wise) | 98.11 | 13.39 | 68.82 | 0.86 | 80.24 | 98.10 | 63.33 | 99.93 | 92.05 | 45.72 | 65.00 | 17.07 | 71.79 | 77.93 | 99.02 | 45.85 |
| Concrete AM (Layer-Wise) | 98.43 | 16.36 | 74.95 | 1.02 | 87.20 | 45.98 | 90.15 | 95.67 | 93.00 | 44.74 | 71.86 | 22.18 | 88.68 | 85.93 | 70.83 | 37.66 |
| ANP | 98.24 | 13.60 | 74.24 | 0.98 | 87.05 | 54.80 | 91.85 | 91.59 | 92.60 | 39.50 | 70.27 | 23.19 | 89.45 | 85.71 | 73.20 | 37.28 |
| AWM | 98.27 | 13.61 | 74.23 | 1.02 | 87.11 | 54.93 | 91.78 | 91.63 | 92.62 | 38.70 | 70.37 | 23.03 | 89.45 | 85.73 | 73.28 | 37.15 |
| SAU | 98.24 | 13.57 | 74.13 | 0.96 | 87.06 | 54.24 | 91.85 | 91.67 | 92.60 | 39.03 | 70.21 | 23.19 | 89.46 | 85.68 | 72.96 | 37.11 |
| **DAM(Ours)** | 98.93 | 14.64 | 69.26 | 0.90 | 88.34 | 43.38 | 91.19 | 85.56 | 92.69 | 47.13 | 70.96 | 22.98 | 89.77 | 85.23 | 64.47 | 35.77 |

Table 3: Results of multi-task merging while adopting four models attacked by TrojVit (CLIP-ViT-B/32, ACC↑/ASR↓). We highlight the best average score in bold and the second score with underline.

| METHOD | MNIST(backdoor) ACC | ASR | Cars(backdoor) ACC | ASR | RESISC45(backdoor) ACC | ASR | EuroSAT(backdoor) ACC | ASR | SVHN(clean) ACC | ASR | DTD(clean) ACC | ASR | AVG ACC(4) | ACC(6) | ASR(4) | ASR(6) |
|---|---|---|---|---|---|---|---|---|---|---|---|---|---|---|---|---|
| Individual(All Clean) | 99.60 | 20.78 | 78.14 | 2.39 | 95.30 | 42.11 | 99.07 | 24.37 | 96.30 | 36.24 | 78.72 | 22.77 | 93.03 | 91.19 | 22.41 | 24.78 |
| Individual(All Backdoor) | 97.47 | 98.56 | 64.00 | 99.63 | 89.00 | 98.37 | 94.85 | 100.00 | 84.86 | 89.67 | 61.50 | 98.90 | 86.33 | 81.95 | 99.14 | 97.52 |
| Individual(Four Backdoor) | 97.47 | 98.56 | 64.00 | 99.63 | 89.00 | 98.37 | 94.85 | 100.00 | 96.30 | 36.24 | 78.72 | 22.77 | 86.33 | 86.72 | 99.14 | 75.93 |
| Weight Average | 91.87 | 68.17 | 62.17 | 53.81 | 74.24 | 80.16 | 82.85 | 100.00 | 67.67 | 89.11 | 52.82 | 22.02 | 77.78 | 71.94 | 75.54 | 68.88 |
| Fisher Merging | 92.88 | 77.04 | 49.87 | 77.95 | 71.70 | 17.13 | 86.11 | 98.22 | 78.68 | 87.30 | 60.59 | 27.18 | 75.14 | 73.31 | 67.59 | 64.14 |
| RegMean | 97.32 | 67.03 | 64.71 | 80.90 | 75.86 | 52.44 | 93.70 | 97.85 | 91.34 | 69.71 | 65.96 | 26.91 | 82.90 | 81.48 | 74.56 | 65.81 |
| Task Arithmetic | 91.90 | 67.86 | 61.99 | 53.25 | 74.27 | 79.65 | 83.11 | 100.00 | 67.49 | 88.96 | 52.87 | 21.60 | 77.82 | 71.94 | 75.19 | 68.55 |
| Ties-Merging | 97.31 | 81.70 | 58.46 | 90.54 | 64.97 | 97.41 | 76.85 | 100.00 | 80.95 | 89.07 | 52.66 | 3.83 | 74.40 | 71.87 | 92.41 | 77.09 |
| AdaMerging(Task-Wise) | 97.58 | 72.63 | 51.65 | 24.65 | 69.90 | 97.79 | 63.48 | 100.00 | 72.95 | 89.43 | 61.82 | 16.70 | 70.65 | 69.56 | 73.77 | 66.87 |
| AdaMerging(Layer-Wise) | 98.30 | 19.66 | 73.62 | 2.15 | 87.24 | 52.32 | 91.26 | 92.26 | 92.45 | 54.34 | 70.74 | 23.44 | 87.61 | 85.60 | 41.60 | 40.70 |
| Concrete TA | 98.39 | 33.94 | 58.13 | 55.07 | 77.40 | 94.97 | 93.89 | 94.97 | 90.95 | 60.98 | 49.89 | 9.10 | 81.95 | 78.11 | 69.74 | 58.17 |
| Concrete AM (Task-Wise) | 97.97 | 31.65 | 50.03 | 23.08 | 81.27 | 97.81 | 64.19 | 99.90 | 92.73 | 58.97 | 69.52 | 14.47 | 73.37 | 75.95 | 63.12 | 54.32 |
| Concrete AM (Layer-Wise) | 98.21 | 25.32 | 73.81 | 2.34 | 88.56 | 61.27 | 93.85 | 89.11 | 92.79 | 68.57 | 72.34 | 22.34 | 88.61 | 86.59 | 44.51 | 44.83 |
| ANP | 98.33 | 19.82 | 74.01 | 2.21 | 87.37 | 52.51 | 91.67 | 92.48 | 92.57 | 54.87 | 70.80 | 23.35 | 87.85 | 85.79 | 41.76 | 40.87 |
| AWM | 98.33 | 19.77 | 73.95 | 2.20 | 87.33 | 52.33 | 91.67 | 92.22 | 92.54 | 54.80 | 70.75 | 23.19 | 87.82 | 85.76 | 41.63 | 40.75 |
| SAU | 98.29 | 19.51 | 74.33 | 2.46 | 87.21 | 39.94 | 91.33 | 93.74 | 92.79 | 59.67 | 71.76 | 21.76 | 87.79 | 85.95 | 38.91 | 39.51 |
| **DAM(Ours)** | 98.47 | 18.53 | 74.39 | 2.39 | 87.13 | 32.26 | 91.04 | 82.63 | 92.62 | 57.64 | 71.49 | 22.92 | 87.76 | 85.86 | 33.95 | 36.06 |

merging experiments adopting two backdoored models can be shown in Appendix C, which can further support our conclusion and verify the effectiveness of our proposed DAM.

*DAM can achieve comparable or better effects in addressing the backdoor issues without additional training compared with post-defense methods.* Through the comparison experiments among Concrete AM (layer-wise), post-defense methods (ANP, AWM, and SAU), and DAM shown in Table 2 and Table 10, we observe that though previous post-defense methods can mitigate the backdoor issues on the SOTA merged model in a way, they are still clearly worse than DAM. This can be attributed to their reliance on high-quality labeled data (Wu et al., 2022), which is usually unrealistic during merging. Additionally, their operated target is a merged model that has converged solely based on performance, making it difficult to achieve a balance between performance and safety. Notably, as the efficiency studies of Table 8 show in the Appendix C.4, these post-defense methods introduce additional training costs but DAM can naturally cope with the backdoor issues in an end-to-end training manner without this constraint.

*DAM can achieve robust results during multi-task merging adapted to the different numbers of backdoored models and different types of backdoors.* First, as shown in Table 3, apart from using two backdoored models during merging, we merge four backdoored and two clean models to achieve a multi-task model. The reported results consistently show that DAM can mitigate the backdoors better, owning 11 scores decrease on the four tasks related to the backdoor models(ASR(4)) and nearly 9 score decrease on the full tasks(ASR(6), compared with previous SOTA merging methods. Simultaneously, the ACC of DAM decreases minorly within 1 score on average which can be accepted in a way. Besides, DAM consistently yields superior results than post-defense methods. Then, we additionally introduce another backdoor attack called BadVit (Yuan et al., 2023) to explore the robustness of DAM. The results of Table 9 illustrate that the backdoor types have

little impact on the effect of DAM, which still can outperform existing multi-task merging methods and post-defense methods. These results can further verify the effectiveness and robustness of DAM.

*DAM can achieve better results than other backdoor-related merging methods that are not designed for multi-task merging scenarios or successfully defend their proposed backdoor attack for model merging.* To distinguish WAG (Arora et al., 2024) and LoRA-as-an-Attack (Liu et al., 2024) from our proposed DAM, as shown in Table 4, for each task related to the backdoored individual finetuned model, we additionally select a clean model for this task during multi-task merging. This corresponds to the open-source community scenario, where exists many models for the same task, and part of them are injected with backdoors but we can not know in advance. We can observe that DAM consistently achieves higher ACC and lower ASR in different settings. For the backdoor attack called BadMering (Arora et al., 2024), we mainly utilize it to attack the merged model of Table 2 and Table 3 adopting the DAM strategy. Combining DAM with BadMerging means we would like to check whether BadMerging can further inject backdoor-related parameters that DAM can not identify, further increasing the ASR badly. The results of Table 5 show that it's difficult to clearly increase the ASR adopting the attack proposed by BadMerging. In other words, DAM can successfully defend this new backdoor attack, further verifying its effectiveness in addressing the backdoor issues.

Table 4: Comparison with other backdoor-defense methods that are not designed for multi-task scenarios.

| SETTINGS | METHODS | ACC↑ | ASR↓ |
|---|---|---|---|
| (2backdoor+2clean)+4 clean | LoRA-as-an-Attack/WAG | 83.81 | 29.51 |
| | **DAM(Ours)** | **85.79** | **24.52** |
| (4backdoor+4clean)+2 clean | LoRA-as-an-Attack/WAG | 81.11 | 33.58 |
| | **DAM(Ours)** | **86.81** | **27.11** |

Table 5: The game between the latest backdoor attack for merging (BadMerging) and our proposed backdoor-defense merging (DAM).

| SETTINGS | METHODS | ACC↑ | ASR↓ |
|---|---|---|---|
| 2 backdoor+4 clean | DAM | 85.23 | 35.77 |
| | DAM+BadMerging | 84.78 | 36.35 |
| 4 backdoor+2 clean | DAM | 85.86 | 36.06 |
| | DAM+BadMerging | 85.44 | 37.11 |

*Two masks have different effects on DAM and collectively promote the merged model with a good trade-off between safety and performance.* The experimental settings are consistent with Table 2 and Table 3. As shown in Eq.3, removing the $M_2$ of DAM means we only focus on the interference of multi-task merging, which is similar to Concrete (Tang et al., 2023). In contrast, removing the $M_2$ of DAM means we mainly deal with the backdoor issues, which can be achieved by adopting AdaMerging (Yang et al., 2023) on the perturbated data. Then, removing two masks together means we only focus on learning the merging coefficients without revising the task vectors. The results shown in Table 6 can collectively verify the effectiveness of the dual-mask optimization of DAM.

Table 6: Ablation studies for the masks of DAM.

| SETTINGS | MASK M1 | MASK M2 | ACC↑ | ASR↓ |
|---|---|---|---|---|
| 2 backdoor+4clean | ✓ | ✓ | 85.23 | **35.77** |
| | × | ✓ | 80.25 | 37.04 |
| | ✓ | × | 85.93 | 37.66 |
| | × | × | 85.47 | 37.08 |
| 4backdoor+2clean | ✓ | ✓ | 85.86 | **36.06** |
| | × | ✓ | 80.15 | 40.12 |
| | ✓ | × | 86.59 | 44.83 |
| | × | × | 85.61 | 40.71 |

## 5 CONCLUSION

This paper conducts extensive experiments to explore the backdoor effect for multi-task merging, uncovering two neglected but important phenomena: backdoor succession and backdoor transfer. To address these challenges, we propose a novel Defense-Aware Merging (DAM) algorithm through dual mask optimization to identify a shared and safety-aware subspace so as to mitigate interference and backdoor issues for multi-task merging. Extensive experiments on several benchmarks can verify the effectiveness and robustness of DAM. Finally, we hope this study can draw attention to the safety issues of model merging across more scenarios.

## ACKNOWLEDGMENTS

We sincerely thank the anonymous reviewers for their valuable feedback, which has helped us polish the paper. This work is supported by STI 2030—Major Projects (No. 2021ZD0201405), Shenzhen Basic Research Project (Natural Science Foundation) Basic Research Key Project (NO. JCYJ20241202124430041), the Major Project in Judicial Research from Supreme People's Court (NO. GFZDKT2024C08-3), and Huawei AI 100 Schools Program.

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

## A  RELATED WORK

### A.1  MULTI-TASK MODEL MERGING

Multitask model merging aims to provide cost-effective parameter fusion strategies to integrate multiple task-specific finetuned models from the shared pre-trained model into a unified one that can handle various tasks (Tang et al., 2024a; Yang et al., 2024a). The well-known strategy to merge multiple task-specific models is by performing element-wise interpolation on weights, such as Weight Average (Wortsman et al., 2022; Kaddour, 2022; Chronopoulou et al., 2023; Sanyal et al., 2023; Lawson & Qureshi, 2024; Jia et al., 2024), Fisher merging (Matena & Raffel, 2022; Ryu et al., 2023; Nathan et al., 2024), RegMean (Jin et al., 2022), and Task Arithmetic (Ilharco et al., 2022a; Zhang et al., 2023; Ni et al., 2023; Ortiz-Jimenez et al., 2024). To further enhance the effectiveness of merging, many efforts have been devoted to addressing interference among tasks and proposing gradient-conflict-based (Yadav et al., 2024), representation-based (Yang et al., 2023; 2024b), routing-based (Zhao et al., 2024a;b; Tang et al., 2024c;b), and subspace-based methods (Tang et al., 2023; Tam et al., 2024; Yu et al., 2024). Unfortunately, existing model merging studies enhance knowledge transfer but neglect adversarial backdoor propagation through parameter fusion, with insufficient analysis of multi-task system vulnerabilities.

### A.2  MODEL MERGING CONSIDERING THE SAFETY

The safety of machine learning algorithms is the key to their widespread applications (Wu et al., 2022). Recently, many researchers have begun to emphasize safety concerns associated with merging scenarios (Yang et al., 2024a). For example, some works adopt the subspace-based and data-aware merging methods

to deal with the misalignment (Yi et al., 2024; Hammoud et al., 2024) and Intellectual Property Protection (Cong et al., 2024) problems for large language models (LLMs). Consistent with our work that focuses on the backdoor issues, WAG (Arora et al., 2024) and LoRA-as-an-Attack (Liu et al., 2024) conducted preliminary exploration under backdoor defense scenarios. However, they only adopt the initial weight average strategy (Wortsman et al., 2022), to cope with the backdoor issues on the same task without more fine-grained designs(e.g. subspace). Besides, BadMerging (Zhang et al., 2024) only focuses on executing effective backdoor attacks to maximize the ASR to break the safeguard for multi-task merging methods, while we mainly consider the defense strategy to minimize the ASR during merging. To the best of our knowledge, our work is the first to conduct extensive experiments to investigate the backdoor effect for multi-task merging scenarios and provide a novel defense-aware merging algorithm to alleviate this problem.

## B    MORE DETAILS ABOUT THE METHOD

### B.1    THE IMPLEMENTATION OF THE MASK PROCESSING.

Ideally, both $M_1$ and $M_2$ should be optimized alone, but to make the solution easy, we let the $M_1 = M_2$ during the implementation. Then, this single mask can be optimized with two different losses as Eq. 11 when given the initial sampling distribution. Exactly, there are many mask sampling techniques to process the neuron or parameters, we adopt the concrete mask strategy (Tang et al., 2023) as Eq. 7 to revise the task vector $\tau_i$, where $\mathbf{m}$ is a $d$-dimensional real vector in $[0,1]^d$ parameterized by logits $\mathbf{x} \in \mathbb{R}^d$ or probabilities $\mathbf{p} = \sigma(\mathbf{x})$, $d$ is the number of parameters in a neural network, $u$ is a random variable sampled from a uniform distribution on the interval $(0,1)$ and $T$ is the temperature parameter to control the steepness of the sigmoid function $\sigma(\cdot)$. Moreover, the processed $\tau_i'$ is further re-scaled to $\tau_i''$ as Eq.8 to avoid the mask being too sparse to keep the base performance (Yu et al., 2024).

$$\mathbf{m} = \sigma \left( \left( \log \frac{u}{1-u} + \log \frac{\sigma(x)}{1-\sigma(x)} \right) / \mathrm{T} \right) \tag{7}$$

$$\tau_i'' = \frac{\tau_i'}{\mathbb{E}_{m \sim \mathbf{m}}[m]} = \frac{\tau_i \odot \mathbf{m}}{\mathbb{E}_{m \sim \mathbf{m}}[m]}. \tag{8}$$

### B.2    THE PSEUDO CODE OF OUR PROPOSED DAM

Based on the implementation of the mask, we provide the pseudo-code of our proposed DAM to learn a shared and safety-aware subspace through learning a mask for task vectors. Exactly, as 11, this mask is affected by two different losses designed for balancing the performance and safety for multi-task model merging.

### B.3    DISCUSSION ABOUT THE PARETO OPTIMAL BALANCE BETWEEN PERFORMANCE AND SAFETY.

*Proof of Theorem 1.* We prove this by contradiction. Assume $\mathcal{F}$ does not exist. Then for any set of masks $\mathbf{M}$, there always exists another set $\mathbf{M}'$ such that both $P(\mathbf{M}') > P(\mathbf{M})$ and $S(\mathbf{M}') > S(\mathbf{M})$. Consider a sequence of mask sets $\{\mathbf{M}_n\}_{n=1}^{\infty}$ where each $\mathbf{M}_{n+1}$ improves upon $\mathbf{M}_n$ in both performance and safety. Due to the bounded nature of $P(\cdot)$ and $S(\cdot)$ (e.g., accuracy and attack success rate are bounded between 0 and 1), this sequence must converge to some limit point $\mathbf{M}^*$. However, by our assumption, there must exist an $\mathbf{M}'$ that improves upon $\mathbf{M}^*$, contradicting the definition of $\mathbf{M}^*$ as the limit point. Therefore, our initial assumption must be false, and $\mathcal{F}$ must exist.

---

**Algorithm 1** Defense-Aware Merging to Acquire a Shared and Safety-Aware Subspace through a Mask across Tasks Vectors. This mask equals the Task-Shared Mask and the Backdoor-Detection Mask when $M_1 = M_2$

---

1: **Input:**
        1. a pre-trained model $f$ parameterized by $\theta_{\text{pre}}$, a set of fine-tuned task vectors $\mathcal{T} = \{\tau_i\}_{i=1}^n$, which are partially injected backdoor,
        2. a set of target tasks $\mathcal{S}^{\text{test}}$ including unlabeled data $\mathcal{D}$,
        3. learning rate $\eta_1, \eta_2, \eta_3$, Hyper-parameters $\alpha$, epochs $E$, $L_1$ norm bound $\xi$
2: **Output:** a mask $\mathbf{m}$ parameterized by logits $\mathbf{x}$.
3: Initialize the logits $\mathbf{x}$ to zeros
4: **for** $e = 1$ to $E$ **do**
5:     Initialize $\Delta = \{\Delta_i\}_{i=1}^n$ to zeros
6:     Mask task vectors $\mathcal{T}$ with $\mathbf{m}$ to get $\mathcal{T}'$, Rescale the masked task vectors $\mathcal{T}'$ to get $\mathcal{T}''$
7:     Initialize merging coefficient $\lambda = \{\lambda_i\}_{i=1}^n$ associated with model merging algorithm
8:     $\theta \leftarrow \text{MergeWeight}(\theta_{\text{pre}}, \mathcal{T}''; \lambda)$
9:     **if** $\lambda$ is optimizable **then**
10:         **for** each task $s_i \in \mathcal{S}^{\text{test}}$ **do**
11:             Sample a batch of unlabeled data $\mathcal{D}_i$ from $s_i$
12:             $l_i \leftarrow \mathcal{L}_i(f(\theta), \mathcal{D}_i)$
13:             $l_i^{perturbation} \leftarrow \mathcal{L}_i(f(\theta), \mathcal{D}_i + \Delta_i)$
14:             Clip $\Delta_i$: $\Delta_i = \Delta_i \times \min(1, \frac{\xi}{\|\Delta_i\|_1})$
15:         **end for**
16:         $\lambda' \leftarrow \lambda - \eta_1 \nabla_\lambda \left( \sum_{i=1}^n l_i \right)$
17:         $\theta \leftarrow \text{MergeWeight}(\theta_{\text{pre}}, \mathcal{T}''; \lambda')$ **// Update the merged model with the updated $\lambda'$**
18:         $\Delta \leftarrow \Delta - \eta_2 \nabla_\Delta \left( \sum_{i=1}^n l_i^{perturbation} \right)$ **// Adversarial Trigger Recovery**
19:     **end if**
20:     **for** each task $s_i \in \mathcal{S}$ **do**
21:         Sample a batch of unlabeled data $\mathcal{D}_i$ from $s_i$
22:         $l_i \leftarrow \mathcal{L}_i(f(\theta), \mathcal{D}_i)$
23:         $l_i^{perturbation} \leftarrow \mathcal{L}_i(f(\theta), \mathcal{D}_i + \Delta_i)$
24:     **end for**
25:     $\mathbf{x} \leftarrow \mathbf{x} - \eta_3 \nabla_\mathbf{x} \left( \sum_{i=1}^n (l_i + \alpha l_i^{perturbation}) \right)$
26: **end for**
27: **Return:** the mask parameterized by logits $\mathbf{x}$.

---

**Convergence analysis.** Here we discuss how the proposed algorithm converges to a Pareto optimal solution, balancing performance and safety. Recall the optimization problem in Eq.(3), which can be rewritten as:

$$\mathcal{L}_{\text{perf}}(M_1) = \sum_{i=1}^n \mathcal{L}\left( \mathcal{A}(\theta_{\text{pre}}, \{M_1 \odot \tau_j\}_{j=1}^n, \lambda^*), D_i \right), \tag{9}$$

$$\mathcal{L}_{\text{safe}}(M_1, M_2) = \sum_{i=1}^n \mathcal{L}\left( \mathcal{A}(\theta_{\text{pre}}, \{M_1 \odot M_2 \odot \tau_j\}_{j=1}^n, \lambda^*), D_i + \Delta_i^* \right), \tag{10}$$

$$\mathcal{L}_{total}(\mathbf{M}) = L_{\text{perf}}(M_1) + \alpha L_{\text{safe}}(M_1, M_2). \tag{11}$$

In a multi-objective optimization problem (MOOP), a common approach is to scalarize the objectives by forming a weighted sum (Schäffler et al., 2002; Désidéri, 2012; Burke et al., 2014). Here, $\mathcal{L}_{total}(\mathbf{M})$ serves as the scalarized objective with normalized weights $(\frac{1}{1+\alpha}, \frac{\alpha}{1+\alpha}) \in \triangle^1$, balancing performance and safety

through the parameter $\alpha$. Assume the loss functions $\mathcal{L}_{\text{perf}}(M_1)$ and $\mathcal{L}_{\text{safe}}(M_1, M_2)$ are continuous and convex. With $\alpha > 0$ and suitable learning rates that satisfy standard conditions (e.g., diminishing step sizes or the Robbins-Monro conditions (Robbins & Monro, 1951)), gradient descent methods converge to stationary points in convex optimization. In scalarized MOOP, minimizing a weighted sum of convex objectives with positive weights yields solutions on the Pareto front. At the stationary point $(M_1^*, M_2^*)$, improving one objective (e.g., decreasing $L_{\text{perf}}$) would necessitate worsening the other (increasing $L_{\text{safe}}$). Thus, the solution $\mathbf{M}^* = (M_1^*, M_2^*)$ is Pareto optimal with respect to performance and safety. Adjusting $\alpha$ changes the weights in the scalarized objective, effectively moving the solution along the Pareto front $\mathcal{F}$.

**Corollary 1** (Performance-safety trade-off control). *The hyper-parameter $\alpha$ in Eq.(3) and (11) controls the position on the Pareto front $\mathcal{F}$, with larger $\alpha$ values favoring safety over performance.*

### B.4 DISCUSSIONS WITH THE ROBUSTNESS OF BACKDOOR ATTACK, FINE-TUNING, AND CONTINUAL LEARNING METHODS

We would like to clarify that our setting is indeed different from your mentioned robustness of backdoor attacks, model fine-tuning, and continual learning methods. The core theme of our work is related to model merging using different models (partial backdoor) rather than one model like your mentioned works.

For the traditional backdoor attack, the model provided by the adversary is the final deployment model. However, for model merging, the adversary only contributes to parts of the models, which are provided for the latter model merging, and the adversary has blind knowledge about how model merging is conducted (Zhang et al., 2024). We are the first to explore the backdoor effect (backdoor succession and backdoor transfer) during model merging and provide a defense-aware merging method to mitigate this issue. Exactly, the object of the Backdoor Detection Mask is the same as existing trigger inversion or synthesis methods (Sun & Kolter, 2023; Dunnett et al., 2024), aiming to find a backdoor trigger inserted into the model. The key difference among them exists in the optimization process and special optimization constraints.

Moreover, some trigger inversion works need to recover the backdoor through an optimization process to flip a support set of clean images into the target class (e.g.smoothinv (Sun & Kolter, 2023)) and other works (e.g. BEAGLE (Dunnett et al., 2024)) propose model backdoor forensics techniques and need a few attack samples as instructions. For our proposed DAM as shown in Table 1, the optimization of the backdoor detection mask only needs unlabeled test data. Simultaneously, as shown in Figure 4, both the Backdoor Detection Mask and the Task-Shared Mask contribute to the whole merging process and they are optimized alternately in an iterative process to develop a merged model that effectively balances performance and safety.

Moreover, model merging has its unique challenges compared with finetuning-based methods and continuous learning methods (Zhu et al., 2024; 2025). *From the perspective of the problem*: Fine-tuning-based methods (Zhu et al., 2023; Huang et al., 2024) directly add perturbations to the final model during fine-tuning, continual learning methods additionally consider the forgetting issues related to backdoor attacks during sequential training (Mi et al., 2023; Abbasi et al., 2024; Guo et al., 2024), but both of them only focus on the optimization of its single model during training. In contrast, model merging should additionally consider the interference from other task-specific models. As shown in Figure 4, masking the backdoor-related parameters will also influence the parameters of task interference. There exists a trade-off between performance and safety due to the conflict of these two parts of parameters, which is special for model merging. *From the perspective of training Data available*: As shown in Table 1, we only have the unlabeled test data for model merging, but for finetuning-based and continual learning methods, they need some labeled data at least, which means the setting of model merging is different and difficult.

**Table 7:** The hyperparameters for backdoored models construction.

|  | Cars | RESISC45 | EuroSAT | SVHN | MNIST | DTD |
|---|---|---|---|---|---|---|
| batch_size | 4 | 2 | 2 | 1 | 8 | 2 |
| poison_rate | 0.1 | 0.1 | 0.1 | 0.1 | 0.1 | 0.1 |
| num_patch | 6 | 9 | 9 | 9 | 9 | 9 |
| patch_size | 16 | 16 | 16 | 16 | 16 | 16 |
| attack_learning_rate | 0.22 | 0.22 | 0.22 | 0.22 | 0.22 | 0.22 |
| train_attack_iters | 250 | 250 | 250 | 250 | 250 | 250 |
| attack_target | fc1 | self_attention | self_attention | fc1 | self_attention | self_attention |

### B.5 DISCUSSIONS ABOUT THE PARAMETER DISENTANGLEMENT FOR TASK VECTORS

To clarify our contribution more clearly, we can disentangle the parameter components for the task vectors into two parts: task-general and task-specific parts. The task-general part represents the common parts for different task-specific models. When merging the parameters of clean and backdoored task-specific models:

(i) If the backdoor is injected in the task-general region on the backdoored model, for model merging(average the model weights from clean models and backdoored models), this backdoor effect can be seen as transferring from backdoored models to clean models. That's why the SVHN's ASR significantly increases after merging with the clean model. It's a unique and special phenomenon of existing model merging, which aims to utilize existing checkpoints to construct a new model without the training data for these checkpoints.

(ii) Moreover, if the backdoor is injected in the task-specific region on the backdoor model, the solution can be seen as similar to traditional backdoor defense works, because we don't need to consider the impact of the backdoored models on other clean models.

Exactly, for defenders, we only have different checkpoints without the knowledge of whether or not and which region they are injected with the backdoor. To solve the (i) and (ii) simultaneously, our proposed DAM design two masks to identify the parameters and reset them to pre-trained weights to solve the problem, with the assumption that the pre-trained model is clean and protected.

## C  MORE EXPERIMENTAL RESULTS

### C.1 BACKDOORED MODELS CONSTRUCTIONS

We construct the backdoored models adopting two well-known vit-specific backdoor attack strategies, including TrojVit Zheng et al. (2023) and BadVit (Yuan et al., 2023). We provide detailed hyperparameters during our experiments as the Table 7 shows. Notably, we only use a few data (10 percent of the full test data) for each task to construct backdoored models. The attack target includes the full connection layer(fc) and self-attention layer. Different from classic CNN-specific backdoor attacks(e.g.BadNets (Gu et al., 2017) and LC (Turner et al., 2019)) as the comparison experiments show in BadMerging (Zhang et al., 2024), We mainly select the vit-specific backdoor attack methods(e.g. TrojVit (Zheng et al., 2023)) to inject the patch-wise backdoor, which has been verified to be especially effective for the vision transformer.

The comparison between clean and backdoor models adopting CLIP-ViT-B/32 and CLIP-ViT-L/14 can be shown in Figure 5 and Figure 6. We can observe that the backdoor models achieve good ACC but high ASR compared with the clean model, which may bring the potential risk to model merging. Similar to TrojVit, we simultaneously report the score of ASR for the clean and backdoor models to clarify the backdoor effect that misguides the model output toward the target class when the inputs contain a specific trigger.

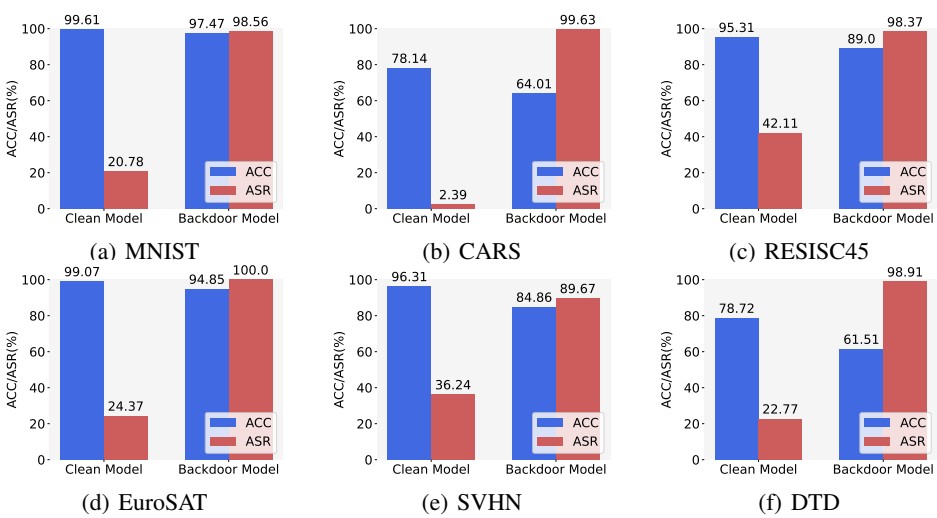

Figure 5: Performance comparison between clean and backdoor(TrojVit) adopting CLIP-ViT-B/32.

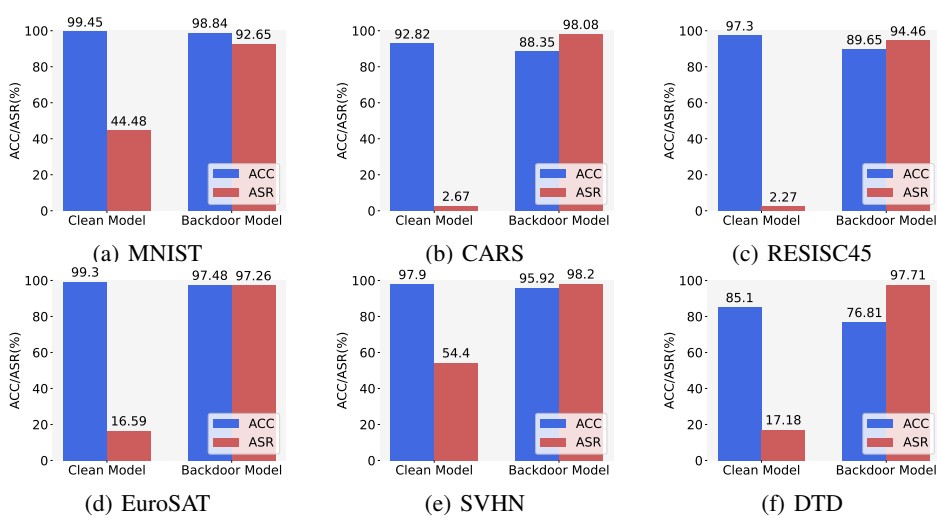

Figure 6: Performance comparison between clean and backdoor(TrojVit) adopting CLIP-ViT-L/14.

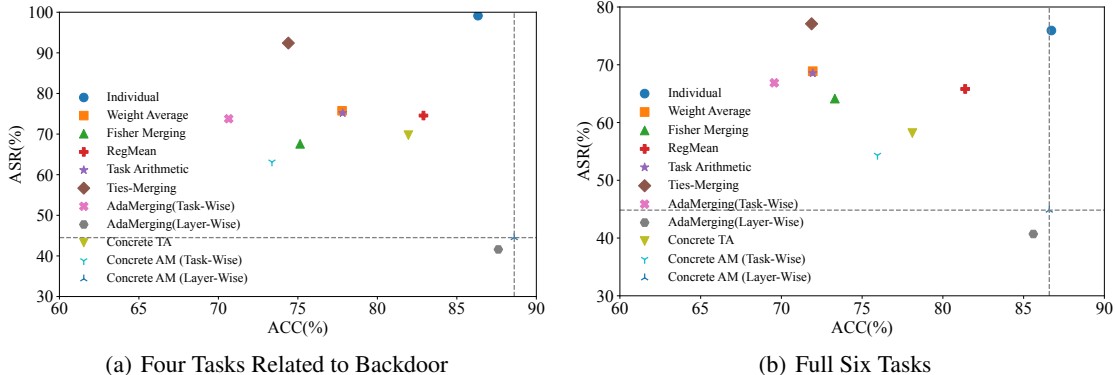

(a) Four Tasks Related to Backdoor        (b) Full Six Tasks

Figure 7: Backdoor Succession Evaluation: Average performance on multi-tasks adopting previous merging methods using four backdoor models (RESISC45, EuroSAT, MNIST, and CARS) and two clean models (SVHN and DTD).

## C.2  MORE EXPERIMENTS ABOUT BACKDOOR SUCCESSION AND BACKDOOR TRANSFER

As shown in Figure 7 and Figure 8, we provide additional experiments while merging four backdoored models and two clean models to further display the phenomenon of backdoor succession and backdoor transfer.

From the results, we can observe that the evaluation of existing merging methods is not fully consistent with the results shown in (Tang et al., 2024a) when we take performance and safety into consideration simultaneously. For example, Ties-Merging achieved unexpectedly poor results, sometimes even worse than RegMean and Fisher Merging, which is the opposite of its reported outcome. This can be attributed to the fact that the success of Ties-Merging relies heavily on the gradient conflict analysis of different tasks. However, this analysis just considers the task-specific performance without dealing with safety issues. When there exists backdoored task-specific models during merging, the causes of gradient conflicts are complex and multidimensional. Notably, compared with task-wise methods, Adamerging(task-wise) and Concrete AM(task-wise), the layer-wise version of corresponding methods can consistently achieve better performance, but their ASR results are still high. This further clarifies the single task-wise performance perspective is not always appropriate, highlighting the need for more exploration of the backdoor issues during multi-task merging.

## C.3  MORE MERGING RESULTS

As shown in Table 10, we provide the merging results when we adopt the CLIP-ViT-L/14 as our pre-trained model and merge two backdoored task-specific models and four clean task-specific models. We can observe that DAM can outperform previous merging methods and post-defense methods in achieving a better trade-off between performance and safety, clearly reducing the ASR without sacrificing the ACC heavily.

Moreover, Table 11 and Table 12 display the merging results while merging two backdoored models and four clean models. Different from Table 2 in the main text, we select other task-specific backdoored models during merging. These results can further verify the effectiveness and robustness of the proposed DAM.

## C.4  EFFICIENCY STUDIES

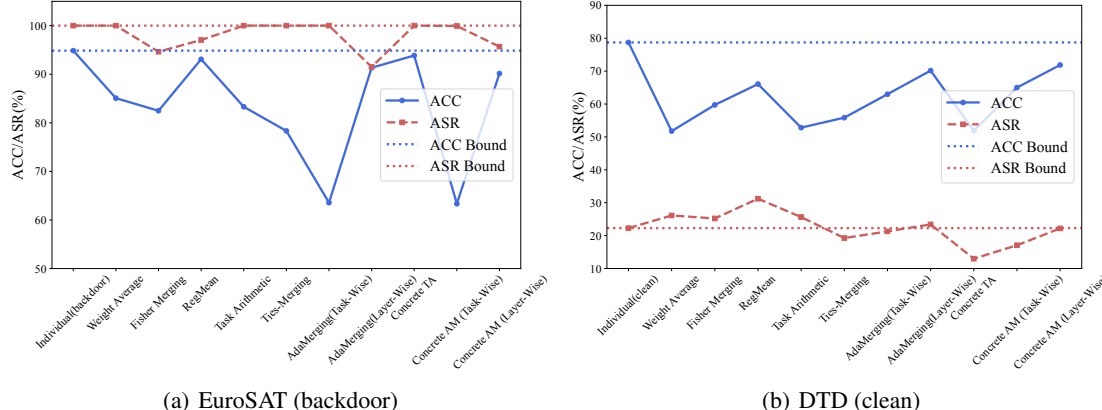

(a) EuroSAT (backdoor)          (b) DTD (clean)

Figure 8: Backdoor Transfer Evaluation: Single-task performance adopting previous merging methods using two backdoor models (RESISC45 and EuroSAT) and four clean models (MNIST, CARS, SVHN and DTD). The ACC Bound and ASR Bound can be set according to the clean or backdoored individual fine-tuned models. The ideal merged model should be close or even upper to the ACC Bound and lower or at least close to the ASR Bound, but different merging methods exhibit unexpected trends due to the backdoor transfer.

Exactly, the reported post-defense methods (AWM and SAU) are two-stage methods (merge first and then defense), and our proposed DAM is an end-to-end merging method without post-hoc cost (consider safety during merging). We can achieve comparable or better performance compared with previous post-defense methods without additional training. We also report the training time to further clarify our contributions, where we calculate the train time (minutes) to achieve the merged model using six-task specific models considering the safety is-

Table 8: Efficiency studies for the comparisons between DAM and post-defense methods.

|  | ANP | AWM | SAU | DAM |
|---|---|---|---|---|
| Merging Time | 29 | 29 | 29 | / |
| Post-Defense time | 15 | 18 | 20 | / |
| Total Time | 44 | 47 | 49 | 35 |

sues on a single Tesla V100 GPU with 32G memory (set the AdamW as the optimizer and the batch size as 16).

### C.5 THE EFFECT OF DOMAIN SOURCE ON MODEL MERGING

It's worthwhile to discuss the domain source of the models used for model merging, which has been neglected by all previous merging methods. To find out the relationship between ASR drop and domain distribution. We first carefully review six used image classification datasets into four categories from the perspective of domain source: (i) Digit images: MNIST and SVHN;(ii) Remote sensing images: RESISC45 and EUROSAT; (iii) Texture images: DTD; (iv) 3D Objects related cars: Stanford cars.

Then, we provide the merging experiments when only the task-specific model on DTD is injected with the backdoor. Other task-specific models are clean and have different domains from the task-specific model on DTD. In this way, we can conduct merging experiments when one backdoor model in a certain domain + several models (backdoored or clean) from different domains. From the results shown in Table 15, we can find that previous merging methods can naturally weaken that backdoor effect through parameter-level merging. But this doesn't verify that our work about defense-aware merging is meaningless. There are two reasons as

**Table 9:** Results of multi-task merging while adopting two models attacked by TrojVit and BadVit respectively (CLIP-ViT-B/32, ACC↑/ASR↓). We highlight the best average score in bold and the second score with underline.

| METHOD | MNIST(clean) ACC | ASR | Cars(clean) ACC | ASR | RESISC45(BadVit) ACC | ASR | EuroSAT(TrojVit) ACC | ASR | SVHN(clean) ACC | ASR | DTD(clean) ACC | ASR | AVG ACC(2) | ACC(6) | ASR(2) | ASR(6) |
|---|---|---|---|---|---|---|---|---|---|---|---|---|---|---|---|---|
| Individual(All Clean) | 99.60 | 20.78 | 78.14 | **2.39** | 95.30 | 42.11 | 99.07 | 24.37 | 96.30 | 36.24 | 78.72 | 22.77 | 97.19 | 91.19 | 33.24 | 24.78 |
| Individual(All Backdoor) | 97.47 | 98.56 | 64.00 | 99.63 | 89.00 | 98.37 | 95.38 | 99.58 | 84.86 | 89.11 | 61.50 | 98.90 | 92.19 | 82.04 | 98.98 | 97.36 |
| Individual(Two Types of Backdoor) | 99.60 | 20.78 | 78.14 | 2.39 | 89.00 | 98.37 | 95.38 | 99.58 | 96.30 | 36.24 | 78.72 | 22.77 | 92.19 | 89.52 | 98.98 | 46.69 |
| Weight Average | 89.87 | 37.23 | 63.44 | 1.51 | 74.51 | 87.33 | 85.44 | 99.31 | 66.99 | 73.89 | 51.22 | 25.11 | 79.98 | 71.91 | 93.32 | 54.06 |
| Fisher Merging | 97.23 | 25.91 | 67.54 | 1.48 | 78.33 | 59.51 | 80.18 | 92.87 | 79.58 | 63.17 | 59.93 | 25.08 | 79.26 | 77.13 | 76.19 | 44.67 |
| RegMean | 98.37 | 19.45 | 69.11 | **2.51** | 73.89 | 58.86 | 91.25 | 90.04 | 92.94 | 42.74 | 61.21 | 29.77 | 82.57 | 81.13 | 74.45 | 40.56 |
| Task Arithmetic | 91.30 | 35.67 | 63.60 | 1.74 | 74.94 | 85.65 | 87.56 | 96.31 | 68.76 | 73.12 | 51.82 | 25.55 | 81.25 | 73.00 | 90.98 | 53.01 |
| Ties-Merging | 98.04 | 16.25 | 63.64 | 0.81 | 66.75 | 98.76 | 76.89 | 97.41 | 84.25 | 49.59 | 52.85 | 19.39 | 71.82 | 73.74 | 98.09 | 47.04 |
| AdaMerging(Task-Wise) | 97.83 | 18.46 | 67.80 | 1.05 | 74.98 | 98.95 | 67.54 | 97.38 | 78.25 | 57.37 | 61.22 | 21.00 | 71.26 | 74.60 | 98.17 | 49.04 |
| AdaMerging(Layer-Wise) | 98.44 | 13.51 | 73.73 | 1.05 | 86.90 | 54.87 | 90.11 | 88.05 | 90.91 | 38.21 | 69.51 | 24.11 | 88.51 | 84.93 | 71.46 | 36.63 |
| Concrete TA | 96.89 | 16.37 | 61.62 | 0.88 | 79.59 | 95.95 | 92.85 | 97.84 | 91.78 | 44.30 | 52.55 | 13.85 | 86.22 | 79.21 | 96.90 | 44.87 |
| Concrete AM (Task-Wise) | 98.68 | 13.39 | 68.82 | 1.11 | 80.24 | 98.10 | 71.25 | 95.93 | 92.78 | 45.72 | 64.58 | 15.41 | 75.75 | 79.39 | 97.02 | 44.94 |
| Concrete AM (Layer-Wise) | 99.11 | 16.36 | 75.95 | 1.09 | 87.20 | 45.98 | 94.51 | 86.25 | 94.14 | 44.74 | 70.33 | 18.51 | **90.86** | **86.87** | 66.12 | 35.49 |
| ANP | 99.00 | 15.58 | 76.58 | 1.33 | 85.55 | 42.51 | 92.57 | 82.51 | 93.58 | 44.04 | 68.98 | 18.35 | 89.06 | 86.04 | 62.51 | 34.05 |
| AWM | 99.11 | 15.63 | 76.42 | 1.18 | 85.48 | 42.38 | 91.77 | 81.01 | 92.99 | 43.81 | 69.25 | 18.68 | 88.63 | 85.84 | 61.70 | 33.78 |
| SAU | 98.69 | 14.87 | 76.02 | 0.98 | 85.87 | 42.98 | 90.59 | 80.58 | 92.21 | 41.58 | 69.65 | 18.78 | 88.23 | 85.51 | 61.78 | 33.30 |
| **DAM(Ours)** | 98.88 | 15.00 | 76.14 | 0.99 | 85.83 | 40.25 | 90.48 | 76.18 | 92.02 | 39.07 | 69.05 | 15.95 | 88.16 | 85.40 | **58.22** | **31.24** |

follows: (i) Take the results on AdaMerging (Layer-Wise) for example, though the ASR on the DTD can decrease badly after merging(98.90->31.33), the ASR on other tasks such as EUROAST(24.37-56.56) can increase unexpectedly. This means apart from the backdoor effect on the task related to the injected model, we should also focus on the ASR on the task related to the clean model used for merging. This new phenomenon during model merging has been explained as the Backdoor Transfer; (ii) Our proposed DAM further lowers the ASR compared with previous merging methods while sacrificing only about 1 in accuracy, achieving the best trade-off between performance and safety.

Moreover, in a way, introducing additional clean models for backdoored models on the same task can be seen as exploring the effect of merging models from the same domain. These experimental results can be also found in Table 5 in the paper. Specifically, during model merging, we only have different checkpoints without the knowledge of whether they are injected with a backdoor or not. Thus, in our paper, it's reasonable to explore whether directly merging the clean models and backdoored model on the same task is enough to mitigate the backdoor as you said. For each task related to the backdoored individual finetuned model, we additionally select a clean model for this task during multi-task merging. Notable, both WAG and LoRA-as-an-Attack defend the backdoor by directly averaging the homogeneous clean and backdoored full model weights or LoRa, we implement them by averaging the weights of original task-specific models and additionally introduced clean models. From the results, we can observe that DAM consistently achieves higher ACC and lower ASR in different settings. These results can verify that the backdoor effect from task-specific models can be mitigated by the clean model from the same domain in a way, but our proposed DAM further achieves higher ACC and lower ASR in different settings.

Table 13: The backdoor transfer evaluated on the SVHN task related to the clean model.

| | ACC(test on SVHN) | ASR(test on SVHN) |
|---|---|---|
| Individual(SVHN_clean) | **96.30** | **36.24** |
| Weight Average | 66.04 | 73.15 |
| Fisher Merging | 78.38 | 62.16 |
| RegMean | 92.94 | 42.74 |
| Task Arithmetic | 67.76 | 73.12 |
| Ties-Merging | 85.06 | 49.59 |
| AdaMerging(Task-Wise) | 77.79 | 57.37 |
| AdaMerging(Layer-Wise) | 92.49 | 38.21 |
| Concrete TA | 91.20 | 44.30 |
| Concrete AM (Task-Wise) | 92.05 | 45.72 |
| Concrete AM (Layer-Wise) | 93.00 | 44.74 |

Table 14: The backdoor succession evaluated on the task EUROSAT related to the backdoored model.

| | ACC(test on EuroSAT) | ASR(test on EuroSAT) |
|---|---|---|
| Individual(EuroSAT_backdoor) | **94.85** | **100.00** |
| Weight Average | 85.07 | 100.00 |
| Fisher Merging | 82.48 | 94.63 |
| RegMean | 93.08 | 97.04 |
| Task Arithmetic | 83.30 | 100.00 |
| Ties-Merging | 78.33 | 100.00 |
| AdaMerging(Task-Wise) | 63.56 | 100.00 |
| AdaMerging(Layer-Wise) | 91.33 | 91.51 |
| Concrete TA | 93.85 | 100.00 |
| Concrete AM (Task-Wise) | 63.33 | 99.93 |
| Concrete AM (Layer-Wise) | 90.15 | 95.67 |

Table 10: Results of multi-task merging while adopting two models attacked by TrojVit (CLIP-ViT-L/14, ACC↑/ASR↓). We highlight the best average score in bold and the second score with underline.

| METHOD | MNIST(clean) | | Cars(clean) | | RESISC45(backdoor) | | EuroSAT(backdoor) | | SVHN(clean) | | DTD(clean) | | AVG | | | |
|---|---|---|---|---|---|---|---|---|---|---|---|---|---|---|---|---|
| | ACC | ASR | ACC | ASR | ACC | ASR | ACC | ASR | ACC | ASR | ACC | ASR | ACC(2) | ACC(6) | ASR(2) | ASR(6) |
| Individual(All Clean) | 99.75 | 44.48 | 92.82 | 2.67 | 97.30 | 2.27 | 99.30 | 16.59 | 97.90 | 54.40 | 85.10 | 17.18 | 98.30 | 95.36 | 9.43 | 22.93 |
| Individual(All Backdoor) | 98.84 | 92.65 | 88.35 | 98.08 | 89.65 | 94.46 | 97.48 | 97.26 | 95.92 | 98.20 | 76.81 | 97.71 | 93.57 | 91.18 | 95.86 | 96.39 |
| Individual(Two Backdoor) | 99.75 | 44.48 | 92.82 | 2.67 | 89.65 | 94.46 | 97.48 | 97.26 | 97.90 | 54.40 | 85.10 | 17.18 | 93.57 | 93.78 | 95.86 | 51.74 |
| Weight Average | 98.05 | 41.55 | 82.63 | 1.88 | 85.46 | 59.92 | 93.52 | 90.70 | 82.26 | 82.69 | 64.04 | 3.99 | 89.49 | 84.33 | 75.31 | 46.79 |
| Fisher Merging | 97.51 | 44.35 | 86.08 | 1.93 | 80.97 | 16.11 | 92.59 | 88.15 | 90.53 | 81.66 | 73.09 | 4.20 | 86.78 | 86.80 | 52.13 | 39.40 |
| RegMean | 99.27 | 33.55 | 89.35 | 1.77 | 90.22 | 50.10 | 96.63 | 66.44 | 96.45 | 77.18 | 76.33 | 8.19 | 93.43 | 91.38 | 58.27 | 39.54 |
| Task Arithmetic | 98.05 | 41.52 | 82.61 | 1.87 | 85.51 | 59.97 | 93.52 | 90.74 | 82.27 | 82.68 | 64.04 | 3.99 | 89.52 | 84.33 | 75.36 | 46.80 |
| Ties-Merging | 99.03 | 25.85 | 84.65 | 1.07 | 86.63 | 94.41 | 89.67 | 94.30 | 92.06 | 74.26 | 65.96 | 2.18 | 88.15 | 86.33 | 94.36 | 48.68 |
| AdaMerging(Task-Wise) | 97.86 | 30.18 | 85.46 | 0.99 | 84.84 | 73.57 | 88.48 | 93.79 | 80.35 | 73.80 | 77.55 | 2.66 | 86.66 | 85.76 | 83.68 | 45.83 |
| AdaMerging(Layer-Wise) | 99.26 | 18.59 | 91.38 | 0.91 | 93.86 | 2.02 | 96.74 | 50.78 | 96.46 | 64.41 | 83.51 | 7.07 | 95.30 | 93.54 | 26.40 | 23.96 |
| Concrete TA | 98.26 | 28.44 | 86.97 | 1.35 | 85.51 | 69.44 | 87.51 | 89.32 | 79.32 | 72.15 | 78.33 | 4.51 | 86.51 | 85.98 | 79.38 | 44.20 |
| Concrete AM (Task-Wise) | 98.86 | 28.18 | 85.87 | 0.99 | 84.51 | 66.57 | 87.44 | 89.54 | 81.10 | 90.27 | 78.21 | 2.99 | 85.98 | 85.54 | 78.06 | 43.01 |
| Concrete AM (Layer-Wise) | 99.06 | 16.89 | 91.64 | 0.93 | 92.88 | 2.12 | 96.44 | 48.25 | 96.57 | 67.51 | 83.51 | 6.88 | 94.66 | 93.35 | 25.19 | 23.76 |
| ANP | 98.45 | 16.55 | 91.69 | 0.91 | 92.00 | 1.98 | 96.77 | 47.55 | 96.81 | 67.59 | 83.00 | 6.54 | 94.39 | 93.12 | 24.77 | 23.52 |
| AWM | 98.15 | 15.89 | 91.33 | 0.77 | 92.14 | 2.12 | 96.40 | 47.40 | 96.22 | 66.12 | 83.11 | 6.12 | 94.27 | 92.89 | 24.76 | 23.07 |
| SAU | 98.00 | 14.79 | 90.89 | 0.54 | 92.18 | 1.85 | 96.32 | 46.40 | 96.00 | 65.32 | 83.18 | 6.21 | 94.25 | 92.76 | 24.13 | 22.52 |
| DAM(Ours) | 97.94 | 14.51 | 91.18 | 0.76 | 92.93 | 2.40 | 96.18 | 43.18 | 92.76 | 55.71 | 82.57 | 4.09 | 94.56 | 92.26 | 22.79 | 20.11 |

Table 11: Results of multi-task merging while adopting two models attacked by TrojVit (CLIP-ViT-B/32, ACC↑/ASR↓). We highlight the best average score in bold and the second score with underline.

| METHOD | MNIST(backdoor) | | Cars(backdoor) | | RESISC45(clean) | | EuroSAT(clean) | | SVHN(clean) | | DTD(clean) | | AVG | | | |
|---|---|---|---|---|---|---|---|---|---|---|---|---|---|---|---|---|
| | ACC | ASR | ACC | ASR | ACC | ASR | ACC | ASR | ACC | ASR | ACC | ASR | ACC(2) | ACC(6) | ASR(2) | ASR(6) |
| Individual(All Clean) | 99.60 | 20.78 | 78.14 | 2.39 | 95.30 | 42.11 | 99.07 | 24.37 | 96.30 | 36.24 | 78.72 | 22.77 | 88.87 | 91.19 | 11.59 | 24.78 |
| Individual(All Backdoor) | 97.47 | 98.56 | 64.00 | 99.63 | 89.00 | 98.37 | 94.85 | 100.00 | 84.86 | 89.67 | 61.50 | 98.90 | 80.74 | 81.95 | 99.10 | 97.52 |
| Individual(Two Backdoor) | 97.47 | 98.56 | 64.00 | 99.63 | 95.30 | 42.11 | 99.07 | 24.37 | 96.30 | 36.24 | 78.72 | 22.77 | 80.74 | 88.48 | 99.10 | 53.95 |
| Weight Average | 91.88 | 68.53 | 62.65 | 50.48 | 73.60 | 15.73 | 83.04 | 85.63 | 67.67 | 89.36 | 52.87 | 23.62 | 77.27 | 71.95 | 59.51 | 55.56 |
| Fisher Merging | 93.61 | 74.73 | 49.68 | 68.24 | 75.22 | 7.51 | 85.19 | 91.56 | 78.24 | 87.25 | 59.68 | 22.82 | 71.65 | 73.60 | 71.49 | 58.69 |
| RegMean | 97.61 | 61.30 | 65.28 | 81.58 | 85.87 | 9.49 | 96.11 | 54.07 | 92.04 | 67.84 | 67.23 | 27.18 | 81.45 | 84.02 | 71.44 | 50.24 |
| Task Arithmetic | 91.84 | 68.15 | 62.49 | 49.98 | 73.63 | 15.33 | 83.22 | 85.70 | 67.50 | 89.22 | 53.09 | 22.98 | 77.17 | 71.96 | 59.07 | 55.23 |
| Ties-Merging | 97.28 | 84.39 | 59.46 | 94.66 | 75.32 | 13.92 | 84.26 | 83.85 | 81.10 | 90.27 | 75.62 | 53.62 | 78.37 | 75.17 | 89.53 | 62.22 |
| AdaMerging(Task-Wise) | 96.60 | 56.58 | 49.73 | 10.58 | 78.95 | 34.22 | 80.11 | 22.71 | 77.67 | 81.37 | 61.65 | 14.79 | 73.17 | 74.12 | 33.58 | 36.71 |
| AdaMerging(Layer-Wise) | 98.24 | 18.97 | 73.88 | 1.80 | 87.11 | 11.95 | 91.52 | 59.22 | 91.71 | 53.14 | 70.48 | 22.71 | 86.06 | 85.67 | 10.39 | 27.97 |
| Concrete TA | 98.41 | 37.07 | 60.24 | 56.56 | 80.00 | 26.44 | 95.15 | 81.59 | 90.96 | 62.27 | 52.50 | 12.39 | 79.33 | 79.54 | 46.82 | 46.05 |
| Concrete AM (Task-Wise) | 97.38 | 25.54 | 52.52 | 10.58 | 85.27 | 36.19 | 92.07 | 64.78 | 93.16 | 69.80 | 53.62 | 13.30 | 74.95 | 81.31 | 18.06 | 32.95 |
| Concrete AM (Layer-Wise) | 98.66 | 20.75 | 75.07 | 2.60 | 89.95 | 13.48 | 94.07 | 52.37 | 93.81 | 60.81 | 71.86 | 19.63 | 86.87 | 87.24 | 11.68 | 28.27 |
| ANP | 97.25 | 19.84 | 74.81 | 1.89 | 90.32 | 12.98 | 95.48 | 58.74 | 93.89 | 59.15 | 70.81 | 19.79 | 86.68 | 87.01 | 11.51 | 28.71 |
| AWM | 98.58 | 22.74 | 74.63 | 2.54 | 90.29 | 14.63 | 95.22 | 41.19 | 92.97 | 63.43 | 71.17 | 18.32 | 86.61 | 87.14 | 12.64 | 27.14 |
| SAU | 98.40 | 23.53 | 73.44 | 2.60 | 90.33 | 11.29 | 94.85 | 38.93 | 91.02 | 65.59 | 71.12 | 17.18 | 85.92 | 86.53 | 13.07 | 26.52 |
| DAM(Ours) | 98.62 | 17.83 | 74.01 | 1.13 | 88.51 | 10.13 | 89.41 | 57.45 | 93.37 | 51.24 | 72.77 | 18.09 | 86.32 | 86.12 | 9.48 | 25.98 |

Table 12: Results of multi-task merging while adopting two models attacked by TrojVit (CLIP-ViT-B/32, ACC↑/ASR↓). We highlight the best average score in bold and the second score with underline.

| METHOD | MNIST(clean) | | Cars(clean) | | RESISC45(clean) | | EuroSAT(clean) | | SVHN(backdoor) | | DTD(backdoor) | | AVG | | | |
|---|---|---|---|---|---|---|---|---|---|---|---|---|---|---|---|---|
| | ACC | ASR | ACC | ASR | ACC | ASR | ACC | ASR | ACC | ASR | ACC | ASR | ACC(2) | ACC(6) | ASR(2) | ASR(6) |
| Individual(All Clean) | 99.60 | 20.78 | 78.14 | 2.39 | 95.30 | 42.11 | 99.07 | 24.37 | 96.30 | 36.24 | 78.72 | 22.77 | 87.51 | 91.19 | 29.51 | 24.78 |
| Individual(All Backdoor) | 97.47 | 98.56 | 64.00 | 99.63 | 89.00 | 98.37 | 94.85 | 100.00 | 84.86 | 89.67 | 61.50 | 98.90 | 73.18 | 81.95 | 94.29 | 97.52 |
| Individual(Two Backdoor) | 99.60 | 20.78 | 78.14 | 2.39 | 95.30 | 42.11 | 99.07 | 24.37 | 84.86 | 89.67 | 61.50 | 98.90 | 73.18 | 86.41 | 94.29 | 46.37 |
| Weight Average | 91.33 | 60.86 | 63.82 | 1.24 | 74.90 | 18.62 | 83.11 | 83.48 | 67.89 | 92.89 | 52.82 | 73.62 | 60.36 | 72.31 | 83.26 | 55.12 |
| Fisher Merging | 90.58 | 42.75 | 66.12 | 1.53 | 79.98 | 15.86 | 92.78 | 83.67 | 83.59 | 82.01 | 52.29 | 31.81 | 67.94 | 77.56 | 56.91 | 42.94 |
| RegMean | 96.16 | 42.10 | 67.90 | 1.46 | 84.29 | 8.63 | 95.93 | 32.22 | 84.24 | 81.81 | 60.64 | 69.31 | 72.44 | 81.53 | 75.56 | 39.26 |
| Task Arithmetic | 89.49 | 60.54 | 63.20 | 1.21 | 73.02 | 18.08 | 85.19 | 83.67 | 65.89 | 92.73 | 51.70 | 72.82 | 58.80 | 71.42 | 82.78 | 54.84 |
| Ties-Merging | 86.67 | 53.70 | 58.95 | 0.31 | 73.63 | 19.27 | 81.19 | 74.96 | 73.44 | 91.43 | 50.37 | 96.49 | 61.91 | 70.71 | 93.96 | 56.03 |
| AdaMerging(Task-Wise) | 98.36 | 33.80 | 65.24 | 0.58 | 82.08 | 49.08 | 84.67 | 86.04 | 58.92 | 84.27 | 43.72 | 71.22 | 51.32 | 72.17 | 77.75 | 54.17 |
| AdaMerging(Layer-Wise) | 97.92 | 16.01 | 73.71 | 0.81 | 87.49 | 12.95 | 90.93 | 60.26 | 92.34 | 62.42 | 70.27 | 28.83 | 81.31 | 85.44 | 45.63 | 30.21 |
| Concrete TA | 98.02 | 22.36 | 61.33 | 0.52 | 79.59 | 32.05 | 95.19 | 77.33 | 88.28 | 63.07 | 51.87 | 74.52 | 70.08 | 79.05 | 68.80 | 44.98 |
| Concrete AM (Task-Wise) | 98.50 | 15.74 | 69.17 | 0.48 | 87.00 | 48.10 | 93.78 | 74.30 | 58.63 | 76.90 | 40.27 | 46.91 | 49.45 | 74.56 | 61.91 | 43.74 |
| Concrete AM (Layer-Wise) | 98.60 | 14.54 | 74.69 | 0.99 | 90.30 | 14.05 | 93.48 | 60.04 | 93.73 | 57.79 | 71.07 | 29.73 | 82.40 | 86.98 | 43.76 | 29.52 |
| ANP | 98.54 | 15.99 | 75.19 | 1.09 | 89.73 | 16.75 | 92.93 | 64.33 | 93.83 | 59.49 | 71.44 | 29.84 | 82.64 | 86.94 | 44.67 | 31.25 |
| AWM | 97.94 | 15.94 | 74.06 | 0.83 | 87.75 | 13.21 | 91.33 | 60.70 | 92.44 | 62.27 | 70.37 | 28.89 | 81.41 | 85.65 | 45.58 | 30.31 |
| SAU | 98.49 | 18.65 | 73.30 | 1.54 | 90.48 | 14.70 | 94.74 | 39.81 | 90.01 | 63.33 | 70.11 | 22.82 | 80.06 | 86.19 | 43.07 | 26.81 |
| DAM(Ours) | 98.48 | 14.77 | 73.92 | 0.80 | 88.97 | 13.16 | 88.56 | 39.45 | 93.24 | 57.64 | 71.01 | 24.15 | 82.13 | 85.70 | 40.90 | 25.00 |

Table 15: Results of multi-task merging for domain exploration while the task-specific model on DTD is injected with the backdoor. We highlight the best average score in bold and the second score with underline.

| METHOD | MNIST(clean) ACC | ASR | Cars(clean) ACC | ASR | RESISC45(clean) ACC | ASR | EuroSAT(clean) ACC | ASR | SVHN(backdoor) ACC | ASR | DTD(backdoor) ACC | ASR | AVG ACC(DTD) | ACC(6) | ASR(DTD) | ASR(6) |
|---|---|---|---|---|---|---|---|---|---|---|---|---|---|---|---|---|
| Individual(DTD_backdoor) | 99.60 | 20.78 | 78.14 | 2.39 | 95.30 | 42.11 | 99.07 | 24.37 | 96.30 | 36.24 | 61.50 | 98.90 | 61.50 | 88.32 | 98.90 | 37.47 |
| Weight Average | 91.41 | 36.79 | 63.16 | 1.26 | 73.06 | 18.32 | 85.07 | 83.60 | 67.44 | 72.00 | 52.07 | 72.40 | 52.07 | 72.04 | 72.40 | 47.40 |
| Fisher Merging | 91.81 | 33.23 | 66.16 | 1.38 | 80.13 | 13.63 | 94.70 | 73.26 | 76.27 | 61.64 | 52.87 | 30.43 | 52.87 | 76.99 | 30.43 | 35.60 |
| RegMean | 98.35 | 18.29 | 67.97 | 1.34 | 83.84 | 10.40 | 95.00 | 33.37 | 92.95 | 41.12 | 60.58 | 69.47 | 60.58 | 83.12 | 69.47 | 29.00 |
| Task Arithmetic | 98.15 | 13.60 | 43.64 | 0.12 | 56.35 | 45.22 | 69.70 | 82.26 | 78.10 | 33.14 | 39.15 | 87.82 | 39.15 | 64.18 | 87.82 | 43.69 |
| Ties-Merging | 97.91 | 15.12 | 60.65 | 0.42 | 73.41 | 22.84 | 82.22 | 74.52 | 85.17 | 46.01 | 50.85 | 95.74 | 50.85 | 75.04 | 95.74 | 42.44 |
| AdaMerging(Task-Wise) | 97.34 | 15.69 | 63.51 | 0.70 | 79.25 | 49.63 | 83.59 | 81.37 | 78.53 | 48.36 | 42.39 | 66.60 | 42.39 | 74.10 | 66.60 | 43.73 |
| AdaMerging(Layer-Wise) | 98.13 | 13.09 | 73.74 | 0.82 | 87.22 | 13.33 | 90.78 | 60.78 | 92.71 | 38.31 | 69.95 | 29.57 | 69.95 | 85.42 | 29.57 | 25.98 |
| Concrete TA | 98.42 | 17.17 | 62.26 | 0.57 | 79.88 | 32.24 | 95.26 | 77.22 | 91.17 | 46.09 | 51.54 | 71.78 | 51.54 | 79.76 | 71.78 | 40.85 |
| Concrete AM (Task-Wise) | 98.19 | 14.33 | 68.45 | 0.71 | 85.78 | 46.05 | 90.19 | 70.15 | 92.33 | 40.58 | 41.70 | 55.05 | 41.70 | 79.44 | 55.05 | 37.81 |
| Concrete AM (Layer-Wise) | 98.68 | 14.51 | 75.44 | 1.04 | 89.68 | 15.90 | 93.74 | 56.56 | 93.90 | 51.32 | 70.90 | 31.33 | **70.90** | **87.06** | 31.33 | 28.44 |
| DAM(Ours) | 98.63 | 14.60 | 75.36 | 1.03 | 89.97 | 13.98 | 93.56 | 55.44 | 92.24 | 34.57 | 70.85 | 24.31 | 70.85 | 86.77 | **24.31** | **23.99** |

