# OpenReview forum: "Mitigating the Backdoor Effect for Multi-Task Model Merging via Safety-Aware Subspace"
_ICLR.cc/2025/Conference — ICLR 2025 Poster_

### Official Review · Reviewer_3WWP · 2024-10-31

**Soundness:** 3
**Presentation:** 3
**Contribution:** 3
**Rating:** 6
**Confidence:** 4

**Summary:**

This paper first studies the impact of backdoors on model merging, and proposes a noval defense method, Defense-Aware Merging, which uses two masks to find the task-shared model subspace (M1) and the backdoor-oriented parameters (M2). The first mask finds shared subspaces to maintain good performance when merging various task-specific models, and the second mask hides sensitive backdoor neurons by introducing perturbations. The authors formed a bi-level optimization to find the best merging coefficients and masks. The performance of the proposed method is evaluated on several SOTA merging methods compared to post-defenses such as ANP, AWM and SAU. The attack strategies are TrojVit and BadVit. The proposed method can mitigate the backdoor effect while maintaining good performance.

**Strengths:**

1) First work to mitigate the backdoor effect on model merging.
2) The paper is well-written and easy to follow.

**Weaknesses:**

- In Figure 2(a), RESISC 45 and EuroSAT are datasets from the same domain, and it is not surprising to see that this merging can achieve high ACC while maintaining high ASR. In Figure 7(a), the ASR drops sharply due to the merging of different task-specifc models. It is curious to see how much the ASR drops when one task-specific model is merged with other different task-specific models. If ASR drops significantly in this case, one can simply merge the suspected model with several clean models to mitigate the backdoor effect while maintaining good ACC. What advantages can such a defense bring us?

- The backdoor succession and backdoor transfer seem to express the same meaning. Please clearify the difference between this two definitions.

- The paper, Wu et al., 2024, is already withdrawed. Please update the related references to stand your point, such as, "Assisted by the synthesized perturbations, we can identify and adjust the parameters related to the backdoor during merging, assuming that the backdoor-related parameters are more sensitive to the perturbations" in Sec 3. Below is an available one.
[1] Wu, B., Chen, H., Zhang, M., Zhu, Z., Wei, S., Yuan, D., & Shen, C. (2022). Backdoorbench: A comprehensive benchmark of backdoor learning. Advances in Neural Information Processing Systems, 35, 10546-10559.

**Questions:**

Summarize from weakneeses:

Q1: Please provide an experiment or analysis comparing the ASR drop when merging models from similar vs different domains. Specifically, one backdoor model in a certain domain + several models (backdoored or clean) from different domain.

Q2: Why learn M1 and M2, not a single M which combines the utility of M1 and M2? Considering Table 7, the ASR increases when DAM only with M2.

---

> ### Author Response · Authors · 2024-11-24
> **Reply to Reviewer 3WWP (1/4)**
>
> We are grateful for your careful and professional feedback. We are glad to address your questions one by one.
>
> ***1、Discussions and Experiments about the domain source of the models used for merging***
>
> >**Question 1**: Please provide an experiment or analysis comparing the ASR drop when merging models from similar vs different domains. Specifically, one backdoor model in a certain domain + several models (backdoored or clean) from different domains.
>
> Thanks for your careful and professional review. It's really a good question to discuss the domain source of the models used for model merging, which has been neglected by all previous merging methods.
>
> To address your concerns about the relationship between ASR drop and domain distribution. We first carefully review six used image classification datasets into four categories from the perspective of domain source: (i) Digit images: MNIST and SVHN;(ii) Remote sensing images: RESISC45 and EUROSAT; (iii) Texture images: DTD; () 3D Objects related cars: Stanford cars.
>
> Then we address your concerns through the following experiments.
>
> **(1) Additional Experiments in Appendix**
>
> Exactly, apart from RESISC45 and EUROSTAT, which come from a similar domain as you said, we have conducted extensive merging experiments that the backdoor models and other clean models come from different domains, including (i) merging when MNIST and Standford cars are backdoor-related models and (ii) merging when SVHN and DTD are backdoor-related models. The detailed results are in Table 9 and Table 10 in the Appendix. These results can further that our DAM can achieve the best trade-off between safety and performance, which can strengthen our contributions.

---

> ### Author Response · Authors · 2024-11-24
> **Reply to Reviewer 3WWP (2/4)**
>
> To better help you understand the contribution of our work, we further conduct experiments during the Rebuttal as follows:
>
> **(2) Merging experiments when one backdoor model in a certain domain + several models (backdoored or clean) from different domains**
>
> We provide the merging experiments when only the task-specific model on DTD is injected with the backdoor. Other task-specific models are clean and have different domains with the task-specific model on DTD, which can meet the condition as you said.
>
> |           METHOD          | MNIST(clean) |        | Cars(clean) |       | RESISC45(clean) |        | EuroSAT(clean) |        | SVHN(clean) |        | DTD(backdoor) |        |   AVG   |         |           |           |
> |:-------------------------:|:------------:|:------:|:-----------:|:-----:|:---------------:|:------:|:--------------:|:------:|:-----------:|:------:|:-------------:|:------:|:-------:|:-------:|:---------:|:---------:|
> |                           |      ACC     |   ASR  |     ACC     |  ASR  |       ACC       |   ASR  |       ACC      |   ASR  |     ACC     |   ASR  |      ACC      |   ASR  | ACC_DTD | ACC_ALL |  ASR_DTD  |  ASR_ALL  |
> |  Individual(DTD_backdoor) |    99.60     | 20.78  |    78.14    | 2.39  |      95.30      | 42.11  |     99.07      | 24.37  |    96.30    | 36.24  |     61.50     | 98.90  |  61.50  |  88.32  |   98.90   |   37.47   |
> |       Weight Average      |    91.41     | 36.79  |    63.16    | 1.26  |      73.06      | 18.32  |     85.07      | 83.60  |    67.44    | 72.00  |     52.07     | 72.40  |  52.07  |  72.04  |   72.40   |   47.40   |
> |       Fisher Merging      |    91.81     | 33.23  |    66.16    | 1.38  |      80.13      | 13.63  |     94.70      | 73.26  |    76.27    | 61.64  |     52.87     | 30.43  |  52.87  |  76.99  |   30.43   |   35.60   |
> |          RegMean          |    98.35     | 18.29  |    67.97    | 1.34  |      83.84      | 10.40  |     95.00      | 33.37  |    92.95    | 41.12  |     60.58     | 69.47  |  60.58  |  83.12  |   69.47   |   29.00   |
> |      Task Arithmetic      |    98.15     | 13.60  |    43.64    | 0.12  |      56.35      | 45.22  |     69.70      | 82.26  |    78.10    | 33.14  |     39.15     | 87.82  |  39.15  |  64.18  |   87.82   |   43.69   |
> |        Ties-Merging       |    97.91     | 15.12  |    60.65    | 0.42  |      73.41      | 22.84  |     82.22      | 74.52  |    85.17    | 46.01  |     50.85     | 95.74  |  50.85  |  75.04  |   95.74   |   42.44   |
> |   AdaMerging(Task-Wise)   |    97.34     | 15.69  |    63.51    | 0.70  |      79.25      | 49.63  |     83.59      | 81.37  |    78.53    | 48.36  |     42.39     | 66.60  |  42.39  |  74.10  |   66.60   |   43.73   |
> |   AdaMerging(Layer-Wise)  |    98.13     | 13.09  |    73.74    | 0.82  |      87.22      | 13.33  |     90.78      | 60.78  |    92.71    | 38.31  |     69.95     | 29.57  |  69.95  |  85.42  |   29.57   |   25.98   |
> |        Concrete TA        |    98.42     | 17.17  |    62.26    | 0.57  |      79.88      | 32.24  |     95.26      | 77.22  |    91.17    | 46.09  |     51.54     | 71.78  |  51.54  |  79.76  |   71.78   |   40.85   |
> |  Concrete AM (Task-Wise)  |    98.19     | 14.33  |    68.45    | 0.71  |      85.78      | 46.05  |     90.19      | 70.15  |    92.33    | 40.58  |     41.70     | 55.05  |  41.70  |  79.44  |   55.05   |   37.81   |
> | Concrete AM  (Layer-Wise) |    98.68     | 14.51  |    75.44    | 1.04  |      89.68      | 15.90  |     93.74      | 56.56  |    93.90    | 51.32  |     70.90     | 31.33  |  70.90  |  87.06  |   31.33   |   28.44   |
> |       **DAM(Ours)**       |    98.63     | 14.60  |    75.36    | 1.03  |      89.97      | 13.98  |     93.56      | 55.44  |    92.24    | 34.57  |     70.85     | 24.31  |  70.85  |  86.77  | **24.31** | **23.99** |
>
> We can find that previous merging methods can naturally weaken that backdoor effect through parameter-level merging. But this doesn't verify that our work about defense-aware merging is meaningless. There are two reasons as follows:
>
> (i) Take the results on AdaMerging (Layer-Wise) for example, though the ASR on the DTD can decrease badly after merging(98.90->31.33), the ASR on other tasks such as EUROAST(24.37-56.56) can increase unexpectedly. This means apart from the backdoor effect on the task related to the injected model, we should also focus on the ASR on the task related to the clean model used for merging. This new phenomenon during model merging has been explained as the **Backdoor Transfer**.
>
> (ii) Our proposed DAM further lowers the ASR compared with previous merging methods while sacrificing only about 1 in accuracy, achieving the best trade-off between performance and safety.

---

> ### Author Response · Authors · 2024-11-24
> **Reply to Reviewer 3WWP (3/4)**
>
> **(3) Merging experiments using the clean and backdoored models for the same task.**
>
> Exactly, in a way, introducing additional clean models for backdoored models on the same task can be seen as exploring the effect of merging models from the same domain. These experimental results can be also found in Table 5 in the paper.
>
> Specifically, during model merging, we only have different checkpoints without the knowledge of whether they are injected with a backdoor or not(lines 200-204). Thus, in our paper, it's reasonable to explore whether directly merging the clean models and backdoored model on the same task is enough to mitigate the backdoor as you said. For each task related to the backdoored individual finetuned model, we additionally select a clean model for this task during multi-task merging. Notable, both WAG and  LoRA-as-an-Attack defend the backdoor by directly averaging the homogeneous clean and backdoored full model weights or LoRa, we implement them by averaging the weights of original task-specific models and additionally introduced clean models.
>
>  The experimental results are shown as follows:
>
> |          SETTINGS          |        METHODS        |    ACC↑   |    ASR↓   |
> |:--------------------------:|:---------------------:|:---------:|:---------:|
> | (2backdoor+2clean)+4 clean | LoRA-as-an-Attack/WAG |   83.81   |   29.51   |
> | (2backdoor+2clean)+4 clean |     **DAM(Ours)**     | **85.79** | **24.52** |
> | (4backdoor+4clean)+2 clean | LoRA-as-an-Attack/WAG |   81.11   |   33.58   |
> | (4backdoor+4clean)+2 clean |     **DAM(Ours)**     | **86.81** | **27.11** |
>
> From the results, we can observe that DAM consistently achieves higher ACC and lower ASR in different settings. These results can verify that the backdoor effect from task-specific models can be mitigated by the clean model from the same domain in a way, but our proposed DAM further achieves higher ACC and lower ASR in different settings.
>
> **Based on the above analysis, we would like to clarify our contribution to answer your question.**
>
> >**Weakness 1**:  “What advantages can such a defense bring us?”
>
> **(i) Point out the drawbacks of the single-aspect evaluation for existing merging methods, which can help reconsider the applicability of previous merging methods.**
>
> To the best of our knowledge, we are the first to explore the backdoor effect for multi-task merging, there exist some unexpected but significant Findings.
>
> For example, Ties-Merging achieved unexpectedly poor results on the safety evaluation metric, sometimes even worse than RegMean and Fisher Merging, which is the opposite of its reported outcome regarding performance. This can be attributed to the fact that the success of Ties-Merging relies heavily on the gradient conflict analysis of different tasks. However, this analysis just considers the task-specific performance without dealing with safety issues. When there exists backdoored task-specific models during merging, the causes of gradient conflicts are complex and multidimensional. Notably, compared with task-wise methods, Adamerging(task-wise) and Concrete AM(task-wise), the layer-wise version of corresponding methods can consistently achieve better performance, but their ASR results are still high. This further clarifies the single task-wise performance perspective is not always appropriate, highlighting the need for more exploration of the backdoor issues during multi-task merging. These contents can be shown in Appendix C.2.
>
> **(ii) Technique Contribution: The better trade-off between the performance and safety of adopting our proposed Defense-Aware merging Method**
>
> The added experiments above show that our proposed DAM still achieves the lowest ASR on the DTD task and the average ASR on the full six tasks under your mentioned setting. Integrated with the superior results of DAM when the models come from the same domain, it can be concluded that our DAM can universally achieve a better trade-off between performance and safety compared with existing merging methods.

---

> ### Author Response · Authors · 2024-11-24
> **Reply to Reviewer 3WWP (4/4)**
>
> ***2、More clarification for the definitions of backdoor succession and backdoor transfer***
>
> >**Weakness 2**: The backdoor succession and backdoor transfer seem to express the same meaning. Please clearify the difference between these two definitions.
>
> Thanks for your questions. The **backdoor succession** describes that the backdoor effect from backdoored task-specific models will succeed during merging even after adopting the SOTA multi-task merging methods.  As shown in Figure 2(a), we provide the two backdoored task-specific models (RESISC45 and EuroSAT) and the other 4 clean models for merging. On RESISC45 and EuroSAT tasks, the ASR value of the merged model is still high. **This shows sometimes merging can not mitigate the backdoor effect from the backdoored models used for merging** To help you understand it more clearly,  we select the  EUROSAT part of Table 2 as follows to clarify our statements. **The ASR on the backdoor-related task (e.g.EUROSAT) decreases but is still high after merging (evaluated on tasks related to the backdoored models)**.
>
> |                                   | ACC(test on EuroSAT) | ASR(test on EuroSAT) |
> |-----------------------------------|:--------------------:|:--------------------:|
> | **Individual(EuroSAT_backdoor))** |       **94.85**      |      **100.00**      |
> |           Weight Average          |        85.07         |        100.00        |
> |           Fisher Merging          |        82.48         |        94.63         |
> |              RegMean              |        93.08         |        97.04         |
> |          Task Arithmetic          |        83.30         |        100.00        |
> |            Ties-Merging           |        78.33         |        100.00        |
> |       AdaMerging(Task-Wise)       |        63.56         |        100.00        |
> |       AdaMerging(Layer-Wise)      |        91.33         |        91.51         |
> |            Concrete TA            |        93.85         |        100.00        |
> |      Concrete AM (Task-Wise)      |        63.33         |        99.93         |
> |     Concrete AM  (Layer-Wise)     |        90.15         |        95.67         |
>
> In contrast,  as shown in Figure 3(b), the **backdoor transfer** describes that though we provide clean task-specific models for merging(e.g. SVHN), the ASR of the merged model on SVHN tasks will increase compared with the individual fine-tuned model on SVHN, due to other backdoored task-specific models(e.g. RESISC45). **This reflects the bad effect caused by the backdoored task-specific model on the clean task-specific model during multi-task merging.** To help you understand it more clearly,  we select the SVHN part of Table 2 as follows to clarify our statements.  **The ASR on the clean task(e.g.SVHN) unexpectedly increases (evaluated on tasks related to the clean models)** due to the model merging using other backdoored models.
>
> |                            | ACC(test on SVHN) | ASR(test on SVHN) |
> |----------------------------|:-----------------:|:-----------------:|
> | **Individual(SVHN_clean)** |     **96.30**     |     **36.24**     |
> |       Weight Average       |       66.04       |       73.15       |
> |       Fisher Merging       |       78.38       |       62.16       |
> |           RegMean          |       92.94       |       42.74       |
> |       Task Arithmetic      |       67.76       |       73.12       |
> |        Ties-Merging        |       85.06       |       49.59       |
> |    AdaMerging(Task-Wise)   |       77.79       |       57.37       |
> |   AdaMerging(Layer-Wise)   |       92.49       |       38.21       |
> |         Concrete TA        |       91.20       |       44.30       |
> |   Concrete AM (Task-Wise)  |       92.05       |       45.72       |
> |  Concrete AM  (Layer-Wise) |       93.00       |       44.74       |
>
> **3、More clarification for the designs of M1 and M2**
>
> > **Question 2**: Why learn M1 and M2, not a single M which combines the utility of M1 and M2?  Considering Table 7, the ASR increases when DAM only with M2.
>
> Exactly, we just utilize a single M which combines the utility of M1 and M2, aiming to solve the optimization more easily. The detailed explanation can be shown in the Algorithm 1 (Lines 764). The separated clarifications of M1 and M2 shown in Figure 4 aim to strengthen our contributions more clearly, simultaneously mitigating the interference issues existing in the task-shared parameters among models and the safety issues existing in the task-specific parameters from the backdoored models.
>
> Moreover, the ablation studies in Table 7 don't show the ASR increase when DAM only with M2 as you said. We are glad to answer your questions if you could provide more details about it
>
> **4、References**
>
> > **Weakness 3**:  The paper, Wu et al., 2024, is already withdrawn. Please update the related references to stand your point
>
> Thanks for your suggestions. We have updated the references you mentioned and carefully check out the full paper.

---

> ### Comment · Reviewer_3WWP · 2024-11-26
>
> Can you explain why the EuroSAT's ASR significantly increases (same phenomenon on SVHN) after merging with the clean model from the perspect of backdoor transfer? Please provide more insights.

---

> ### Author Response · Authors · 2024-11-27
> **Reply to Reviewer 3WWP**
>
> Thanks again for your question about the **backdoor transfer**.
>
> >Can you explain why the EuroSAT's ASR significantly increases (same phenomenon on SVHN) after merging with the clean model from the perspect of backdoor transfer? Please provide more insights.
>
> To clarify this problem, we can disentangle the parameter components for the task vectors into two parts: task-general and task-specific parts. The task-general part represents the common parts for different task-specific models. When merging the parameters of clean and backdoored task-specific models:
>
> (i) If the backdoor is injected in the task-general region on the backdoored model, for model merging(average the model weights from clean models and backdoored models), this backdoor effect can be seen as transferring from backdoored models to clean models. That's why the SVHN's ASR significantly increases after merging with the clean model. It's a unique and special phenomenon of existing model merging, which aims to utilize existing checkpoints to construct a new model without the training data for these checkpoints.
>
> (ii) Moreover, if the backdoor is injected in the task-specific region on the backdoor model, the solution can be seen as similar to traditional backdoor defense works, because we don't need to consider the impact of the backdoored models on other clean models.
>
> Exactly, for defenders, we only have different checkpoints without the knowledge of whether or not and which region they are injected with the backdoor. To solve the (i) and (ii) simultaneously, our proposed DAM design two masks to identify the parameters and reset them to pre-trained weights to solve the problem, with the assumption that the pre-trained model is clean and protected.
>
> We hope our statements can address your concerns. If you have other questions, don't hesitate to tell me. Your suggestions are valuable for us to polish our work and we sincerely hope you can reconsider the score if the mentioned problems have been solved.

---

> > ### Comment · Reviewer_3WWP · 2024-11-27
> >
> > Thanks for your reply. I have changed my rating. Moreover, I suggest this insight can be presented in Sec. 2 to enhance your findings.

---

> > > ### Author Response · Authors · 2024-11-27
> > > **Reply to Reviewer 3WWP**
> > >
> > > Thanks again for your insightful feedback! We will carefully organize the content of the rebuttal in accordance with your suggestions and upload the new PDF as soon as possible.

---

### Official Review · Reviewer_1tZq · 2024-11-04

**Soundness:** 3
**Presentation:** 3
**Contribution:** 3
**Rating:** 6
**Confidence:** 2

**Summary:**

The authors investigate backdoor attacks in multi-task model merging and propose a meta-learning-based optimization method to mitigate these threats. Extensive experiments demonstrate its effectiveness and efficiency.

**Strengths:**

1. Multi-task merging methods are hot and meaningful directions to investigate
2. Meta-learning looks like a reasonable way to consider balancing effectiveness and efficiency, while it is interesting to consider different objectives for outer and inner level.

**Weaknesses:**

1. Bi-level optimization usually computational expensive, the authors might want to justify the sample efficiency and empirically showcase the algorithm running time.
2. I'm not an expert in Vision Transformers, so I'll leave it to the other reviewers to determine if the current experiments are sufficient. However, I am curious whether each task would exhibit different performance levels when subjected to the same backdoor attack. If there are variations, it would be necessary to test each task individually.

**Questions:**

1. Why performance and safety terms can be linearly combined without tradeoff coefficient?
2. Why backdoor trigger and target labels are known during merging in eq (2)?
3. Following Q2, what if the synthesized perturbations are incorrect or inaccurate? Can the method still effective with rough knowledge of backdoor attack?

---

> ### Author Response · Authors · 2024-11-22
> **Reply to Reviewer o55R (1/2)**
>
> We are grateful for your thoughtful feedback. Your assessment is valuable for us to polish our paper. We are glad to address your questions one by one.
>
> ***1、 Efficiency Studies about Bi-Level Optimization***
>
> >**Weakness 1:** Bi-level optimization is usually computationally expensive, the authors might want to justify the sample efficiency and empirically showcase the algorithm running time.
>
> Thanks for your questions. We acknowledge that bi-level optimization is usually computationally expensive. But for merging experiments, we just utilize existing checkpoints of the fine-tuned model without tuning them, the optimization only includes the learning of the mask and merging coefficient in a test-time training manner (as Eq.4). This means the iteration steps are fewer than traditional bi-level optimization.
>
>
> We also report the training time to further clarify our contributions, where we calculate the train time (minutes) to achieve the merged model considering safety using six task-specific models on a single Tesla V100 GPU with 32G memory (set the AdamW as the optimizer and the batch size as 16). Exactly, the reported post-defense methods **(ANP, AWM, and SAU) are two-stage methods (merge first and then defense)**, and **our proposed DAM is an end-to-end merging method without post-hoc cost** (consider safety during merging). The detailed statements can be found in Table 1 and Table 8.
>
> |                       | **ANP** | **AWM** | **SAU** | **DAM(Ours)** |
> |:---------------------:|:-------:|:-------:|:-------:|:-------------:|
> |    **Merging Time**   |    29   |    29   |    29   |       35       |
> | **Post-Defense time** |    15   |    18   |    20   |       /       |
> |     **Total Time**    |    44   |    47   |    49   |       35      |
>
> From the results, we can conclude that DAM can achieve a comparable time cost to other two-stage training baselines involving adversarial training (ANP, AWM, and SAU), which also considers safety issues (as shown in Table 1).
>
> ***2、More Clarification for Experiments***
>
> >**Weakness 2**: I'm not an expert in Vision Transformers, so I'll leave it to the other reviewers to determine if the current experiments are sufficient. However, I am curious whether each task would exhibit different performance levels when subjected to the same backdoor attack. If there are variations, it would be necessary to test each task individually.
>
> Thanks for your questions. Each task would indeed exhibit different performance levels when subjected to the same backdoor attack. In our paper, **we have tested each task individually as you said.** Moreover, we have also released the trigger information of each task for further evaluation in the supplemental materials.
>
> ***3、More Clarification for Method***
>
> > **Question 1**: Why performance and safety terms can be linearly combined without a trade-off coefficient?
>
> Thanks for your questions. We indeed missed the trade-off coefficient in Eq.2 due to carelessness. We have added it in the revised version referring to your questions. But exactly, this does not diminish the contributions of our work. Compared with Eq.1, we just would like to clarify that previous merging works are only designed for good performance of the merged model, we are the first merging work that integrates the performance and safety into the optimization object.
>
> >**Question 2**: Why backdoor trigger and target labels are known during merging in eq (2)?
>
> We are glad to answer your questions. But Eq.2 just clarifies the optimization objects from the perspective of evaluation metrics, maximizing the performance and safety of the merged model.
>
> Exactly, we utilize the test-time adaptation for optimization as shown in Eq.4, which means the test data are set without true labels by default during training. The triggers are only used in the evaluation stage, and during the training stage, they are just approximated as a unified and synthesized perturbation as Eq.3. The specific optimization process of the training stage can refer to Eq.3 and Eq.4.

---

> ### Author Response · Authors · 2024-11-22
> **Reply to Reviewer o55R (2/2)**
>
> >**Question 3: what if the synthesized perturbations are incorrect or inaccurate? Can the method still effective with rough knowledge of backdoor attack?**
>
> **Theory:** The unknown backdoor injections can still be successfully approximated as a unified and synthesized perturbation from the embedding drift theory. Different types of backdoor triggers induce relatively uniform drifts in the model’s embedding space regardless of the trigger location or attack mechanism, we can synthesize a unified perturbation to represent the misguided behavior change upon trigger insertion for each task-specific model. These contents can be found in lines 317-328.
>
> **Experiments:** We conduct experiments adopting **different backdoor injections (Trojvit and BadVit)**, the superior performance of DAM can still achieve the best performance, which can further verify it's reasonable for using the synthesized perturbations to identify different types of triggers. These contents can be also found in Table 4 of the paper.
>
> |           METHOD           | MNIST(clean) |        | Cars(clean) |       | RESISC45(BadVit) |        | EuroSAT(TrojVit) |        | SVHN(clean) |        | DTD(clean) |        |   AVG  |        |           |           |
> |:--------------------------:|:------------:|:------:|:-----------:|:-----:|:----------------:|:------:|:----------------:|:------:|:-----------:|:------:|:----------:|:------:|:------:|:------:|:---------:|:---------:|
> |                            |      ACC     |   ASR  |     ACC     |  ASR  |        ACC       |   ASR  |        ACC       |   ASR  |     ACC     |   ASR  |     ACC    |   ASR  | ACC(2) | ACC(6) |   ASR(2)  |   ASR(6)  |
> |  Individual(two backdoor)  |    99.60     | 20.78  |    78.14    | 2.39  |      89.00       | 98.37  |      95.38       | 99.58  |    96.30    | 36.24  |   78.72    | 22.77  | 92.19  | 89.52  |   98.98   |   46.69   |
> |       Weight Average       |    89.87     | 37.23  |    63.44    | 1.51  |      74.51       | 87.33  |      85.44       | 99.31  |    66.99    | 73.89  |   51.22    | 25.11  | 79.98  | 71.91  |   93.32   |   54.06   |
> |       Fisher Merging       |    97.23     | 25.91  |    67.54    | 1.48  |      78.33       | 59.51  |      80.18       | 92.87  |    79.58    | 63.17  |   59.93    | 25.08  | 79.26  | 77.13  |   76.19   |   44.67   |
> |           RegMean          |    98.37     | 19.45  |    69.11    | 2.51  |      73.89       | 58.86  |      91.25       | 90.04  |    92.94    | 42.74  |   61.21    | 29.77  | 82.57  | 81.13  |   74.45   |   40.56   |
> |       Task Arithmetic      |    91.30     | 35.67  |    63.60    | 1.74  |      74.94       | 85.65  |      87.56       | 96.31  |    68.76    | 73.12  |   51.82    | 25.55  | 81.25  | 73.00  |   90.98   |   53.01   |
> |        Ties-Merging        |    98.04     | 16.25  |    63.64    | 0.81  |      66.75       | 98.76  |      76.89       | 97.41  |    84.25    | 49.59  |   52.85    | 19.39  | 71.82  | 73.74  |   98.09   |   47.04   |
> |    AdaMerging(Task-Wise)   |    97.83     | 18.46  |    67.80    | 1.05  |      74.98       | 98.95  |      67.54       | 97.38  |    78.25    | 57.37  |   61.22    | 21.00  | 71.26  | 74.60  |   98.17   |   49.04   |
> |   AdaMerging(Layer-Wise)   |    98.44     | 13.51  |    73.73    | 1.05  |      86.90       | 54.87  |      90.11       | 88.05  |    90.91    | 38.21  |   69.51    | 24.11  | 88.51  | 84.93  |   71.46   |   36.63   |
> |         Concrete TA        |    96.89     | 16.37  |    61.62    | 0.88  |      79.59       | 95.95  |      92.85       | 97.84  |    91.78    | 44.30  |   52.55    | 13.85  | 86.22  | 79.21  |   96.90   |   44.87   |
> |   Concrete AM (Task-Wise)  |    98.68     | 13.39  |    68.82    | 1.11  |      80.24       | 98.10  |      71.25       | 95.93  |    92.78    | 45.72  |   64.58    | 15.41  | 75.75  | 79.39  |   97.02   |   44.94   |
> |  Concrete AM  (Layer-Wise) |    99.11     | 16.36  |    75.95    | 1.09  |      87.20       | 45.98  |      94.51       | 86.25  |    94.14    | 44.74  |   70.33    | 18.51  | 90.86  | 86.87  |   66.12   |   35.49   |
> | ANP |    99.00     | 15.58  |    76.58    | 1.33  |      85.55       | 42.51  |      92.57       | 82.51  |    93.58    | 44.04  |   68.98    | 18.35  | 89.06  | 86.04  |   62.51   |   34.05   |
> | AWM |    99.11     | 15.63  |    76.42    | 1.18  |      85.48       | 42.38  |      91.77       | 81.01  |    92.99    | 43.81  |   69.25    | 18.68  | 88.63  | 85.84  |   61.70   |   33.78   |
> | SAU |    98.69     | 14.87  |    76.02    | 0.98  |      85.87       | 42.98  |      90.59       | 80.58  |    92.21    | 41.58  |   69.65    | 18.78  | 88.23  | 85.51  |   61.78   |   33.30   |
> |        **DAM(Ours)**       |    98.88     | 15.00  |    76.14    | 0.99  |      85.83       | 40.25  |      90.48       | 76.18  |    92.02    | 39.07  |   69.05    | 15.95  | 88.16  | 85.40  | **58.22** | **31.24** |

---

> > ### Author Response · Authors · 2024-12-01
> > **Waiting for discussions with Reviewer o55R**
> >
> > Dear Reviewer o55R,
> >
> > We sincerely appreciate your valuable suggestions to help us polish our paper. We have carefully considered each point and addressed them one by one in our rebuttal. The updated PDF has been uploaded.
> >
> > As the Author-Review Discussion period draws to a close, we would like to ensure that we have thoroughly addressed all of your concerns and resolved any outstanding issues. We remain open to any further comments or suggestions you may have and are willing to provide additional clarifications if necessary.
> >
> > Of course, we sincerely hope you can reconsider the score and the confidence if your mentioned problems have been solved.
> >
> > Best regards,
> >
> > The Authors

---

### Official Review · Reviewer_XodB · 2024-11-04

**Soundness:** 2
**Presentation:** 2
**Contribution:** 2
**Rating:** 8
**Confidence:** 2

**Summary:**

This paper investigates the vulnerabilities of model merging methods when exposed to backdoor attacks. The paper identifies two primary challenges: 1) backdoor succession, where backdoors remain after merging and 2) backdoor transfer, where backdoors are transferred to clean models. Then the paper proposes Defense-Aware Merging (DAM) to mitigate both task interference and backdoor vulnerabilities. The DAM approach identifies a shared and safety-aware subspace for model merging and optimizing the task-shared mask and backdoor detection mask via bi-level optimization. The experimental results show the effectiveness of DAM in achieving high accuracy and low attack success rates.

**Strengths:**

1. This paper presents the first work to reveal backdoor attack vulnerabilities in existing model merging methods and observes two interesting findings - backdoor succession and backdoor transfer.
2. The paper conducts comprehensive experiments, including the performance of a large number of existing merging models and the comparison of the proposed DAM model with existing backdoor defenses.

**Weaknesses:**

1.	The threat model needs to be further clarified. Does the attacker know the other model? In particular, how does the backdoored model affect other clean models? What are the knowledge and capabilities of the defender? A detailed description of the threat model is missing.
2.	The novel of the proposed DAM approach is incremental, mainly focusing on optimizing two tasks – model merging and backdoor defense. In addition, the defense gain of DAM is marginal compared with existing backdoor defenses, such as SAU and AWM.
3.	The paper investigates the vulnerabilities of model merging methods. The idea is similar to the robustness of backdoor attacks in model fine-tuning and continual learning. It would be great if the paper could compare these topics and discuss the major distinctions and unique challenges in model merging.
4.	It would be great if the paper could provide a brief introduction regarding backdoor attacks mentioned in the paper, such as BadVit, TrojVit, LoRA-as-an-Attack, WAG. In addition, the paper mentioned that BadMerging breaks the safeguard for existing methods, but it is unclear what is the existing safeguard.
5.	The paper writing needs to be improved. Some concepts need further clarification. See my questions below.

**Questions:**

1.	How does the Backdoor Detection Mask differ from existing trigger inversion or synthesis methods?
2.	How does the paper combine DAM with BadMerging as shown in Table 6?
3.	In Figure 4, what is meant by the term “subspace”?
4.	What is LoRA-as-an-Attack/WAG in Table 5?

---

> ### Author Response · Authors · 2024-11-22
> **Reply to Reviewer XodB (1/4)**
>
> Thanks for your careful review. We are glad to address your questions one by one.
>
> **First of all, we would like to re-clarify the multitask merging settings related to the backdoor.**
>
> Multi-task model merging aims to integrate parameters from multiple single-task models finetuned from a common pre-trained model into a unified one to achieve good performance on multi-tasks (lines 37-40). This means **we can directly utilize existing checkpoints to construct a new model without the training data for these checkpoints**.
>
> Thus, under this open-source model ecosystem (lines 46-48), the attackers can download a pre-train model and then inject the model with a backdoor during fine-tuning, and then reload the model to the platform. But for users, they do not know whether that the selected models are injected with a backdoor or not. While conducting multi-task merging experiments adopting these backdoored models, it's necessary to consider the safety issues, which have been neglected by previous merging-related works. These discussions can be also found in lines 222-230, where we clarify the challenges we faced.

---

> ### Author Response · Authors · 2024-11-22
> **Reply to Reviewer XodB (2/4)**
>
> ***1、More Clarification for the Settings and Concepts.***
>
> **(i) The Setting of the Threat Model**
>
> > **Weakness 1**: Does the attacker know the other model? What are the knowledge and capabilities of the defender?
>
> In our work, we assume that adversaries don't know other task-specific models and merging algorithms. For defenders, we only have different checkpoints without the knowledge of whether they are injected with backdoor or not. These contents can be found in Section 2.2 in the revised paper
>
> Exactly, there is no need for adversaries to know other task-specific models. They just have their own poison dataset, fine-tune the pre-trained model on it to achieve a backdoored task-specific model, and then release this trained model for model merging. As shown in Section 2,  due to the backdoor succession and backdoor transfer, the involved backdoored models will influence the safety of the merged model. Moreover, we have conducted extensive experiments to verify the robustness of methods (number of attacks, configuration of poisoned and clean models). In a way, it can correspond to the different settings (e.g. different types of attacks and different understanding levels of other models (execute attack or not)
>
> > **Weakness 1** "How does the backdoored model affect other clean models"
>
> This can be explained as the parameter-level impact when backdoored and clean task-specific models are merged into one model(defined as backdoor transfer in the paper).
>
> **Experimental Perspective**: When backdoored and clean task-specific models are merged into a multi-task one, we can compare the ACC and ASR between the merged model and individual task-specific finetuned models to explore the backdoor effect. As shown in Table 2, we provide the clean task-specific model to merge (low ASR on these tasks: MNIST, CARS, SVHN, and DTD), but the ASR of the merged model on these tasks increases badly. Specifically, we select the SVHN part of Table 2 as follows to clarify our statements. From the results, we can find that the merged model adopting weight average on SVHN increased from 36 to 73 compared with the individual fine-tuned model, even though we provided the clean task-specific model (SVHN clean) for model merging.  These contents can be found in lines 205-214.
>
> |                            |    ACC    |    ASR    |
> |----------------------------|:---------:|:---------:|
> | **Individual(SVHN_clean)** | **96.30** | **36.24** |
> |       Weight Average       |   66.04   |   73.15   |
> |       Fisher Merging       |   78.38   |   62.16   |
> |           RegMean          |   92.94   |   42.74   |
> |       Task Arithmetic      |   67.76   |   73.12   |
> |        Ties-Merging        |   85.06   |   49.59   |
> |    AdaMerging(Task-Wise)   |   77.79   |   57.37   |
> |   AdaMerging(Layer-Wise)   |   92.49   |   38.21   |
> |         Concrete TA        |   91.20   |   44.30   |
> |   Concrete AM (Task-Wise)  |   92.05   |   45.72   |
> |  Concrete AM  (Layer-Wise) |   93.00   |   44.74   |
>
> **Theory Perspective**: For model merging, we can disentangle the parameter components for the task vectors(objects for merging) into two parts: task-general and task-specific parts. The task-general part represents the common parts for different task-specific models. When merging the parameters of clean and backdoored task-specific models:
>
> If the backdoor is injected in the task-general region on the backdoored model, for model merging (average the model weights from clean models and backdoored models), this backdoor effect can be seen as transferring from backdoored models to clean models. That's why the SVHN's ASR significantly increases after merging with the clean model. It's a unique and special phenomenon of existing model merging, which aims to utilize existing checkpoints to construct a new model without the training data for these checkpoints. In contrast, If the backdoor is injected in the task-specific region on the backdoor model, the solution can be seen as similar to traditional backdoor defense works, because we don't need to consider the impact of the backdoored models on other clean models.
>
> Exactly, for defenders, we only have different checkpoints without the knowledge of whether or not and which region they are injected with the backdoor. To solve the (i) and (ii) simultaneously, our proposed DAM design two masks to identify the parameters and reset them to pre-trained weights to solve the problem, with the assumption that the pre-trained model is clean and protected.

---

> ### Author Response · Authors · 2024-11-22
> **Reply to Reviewer XodB (3/4)**
>
> **(ii) Clarification for other backdoor attacks**
>
> > **Weakness 4,5**: It would be great if the paper could provide a brief introduction regarding backdoor attacks mentioned in the paper, such as BadVit, TrojVit, LoRA-as-an-Attack, WAG.
>
> >**BadVit and TrojVit**
>
>  Both BadVit and TrojVit are two commonly used vit-specific backdoor attacks. Compared with CNN-specific backdoors (e.g. Badnets), it executes more fine-grained patch-wise backdoor injection to achieve higher ASR on the vision transformer. These contents can be found in lines 866-872.
>
> >**LoRA-as-an-Attack/WAG**
>
> Both WAG and LoRA-as-an-Attack are merging methods that conducted preliminary exploration under backdoor defense scenarios. However, they just directly average the full model weight or LoRa weights to cope with the backdoor issues on the same task while we focus on the multi-task merging settings with more fine-grained designs(e.g. subspace).  These contents can be found in lines 347-351.
>
>
> >**BadMerging: How does the paper combine DAM with BadMerging as shown in Table 6?**
>
> BadMerging is a newly proposed attack method adapted to the model merging process, which strengthens that adversaries only contribute to parts of models for merging, and have blind knowledge about how model merging is conducted.  These contents can be found in lines 421-436.
>
>  We mainly utilize it to attack the merged process of Table 2 and Table 3 adopting the DAM strategy (2 and 4 backdoor models involved during merging). This means we focus on the game between the trigger optimization of BadMerging for adversaries to inject a backdoor (high ACC and high ASR) and perturbation optimization of our proposed defense-aware merging (DAM) to help detect the backdoor-related parameters and mask them to mitigate the backdoor during merging (high ACC and low ASR).
>
> Thus, combining DAM with BadMerging means we would like to check whether BadMerging can further inject backdoor-related parameters that DAM can not identify, further increasing the ASR badly. But as shown in Experiments of Table 6,  it's difficult to clearly increase the ASR adopting the attack proposed by BadMerging. In other words, DAM can successfully defend this new backdoor attack, further verifying its effectiveness in addressing the backdoor issues. These contents can be found in lines 436-438.
>
>
> >**The term "subspace" In Figure 4**
>
> **The subspace represents the low-dimensional-parameter (part of the full parameters) area compared with full parameters.**
>
> The task vectors (full parameters without mask) of Figure 4 correspond to the optimization directions of different task-specific models, which can be calculated by the difference between pre-trained weights and finetuned model weights.  However, directly merging these task vectors (full parameters without mask) will lead to a merged model with poor performance and safety. As shown in Figure 4, for multi-task merging considering the backdoor issues, the core challenges include the interference existing in the task-shared common parameter area (subspace masking 1) and backdoor-related issues in task-specific parameters (subspace masking 2). Thus, we model mask optimization from the perspective of subspace compared with the full parameters of task vectors, aiming to create a merged model with good performance and safety.
>
> ***2、More Clarification for the Technique Contribution***
>
> **(i) Compared with existing backdoor defense method**
>
> >**Weakness 2**: The novel of the proposed DAM approach is incremental, mainly focusing on optimizing two tasks – model merging and backdoor defense. In addition, the defense gain of DAM is marginal compared with existing backdoor defenses, such as SAU and AWM.
>
> Exactly, the reported post-defense methods **(AWM and SAU) are two-stage methods (merge first and then defense)**, and **our proposed DAM is an end-to-end merging method without post-hoc cost** (consider safety during merging). The detailed statements can be found in lines 399 and Table 1.
>
> Our contribution: (i) Compared with previous merging methods, we are the first to consider safety during merging; (ii) We can achieve comparable or better performance compared with previous post-defense methods without additional training. We also report the training time to further clarify our contributions, where we calculate the train time (minutes) to achieve the merged model using six-task specific models considering the safety issues on a single Tesla V100 GPU with 32G memory (set the AdamW as the optimizer and the batch size as 16).
>
> |                       | **ANP** | **AWM** | **SAU** | **DAM(Ours)** |
> |:---------------------:|:-------:|:-------:|:-------:|:-------------:|
> |    **Merging Time**   |    29   |    29   |    29   |       35       |
> | **Post-Defense time** |    15   |    18   |    20   |       /       |
> |     **Total Time**    |    44   |    47   |    49   |       35      |

---

> ### Author Response · Authors · 2024-11-22
> **Reply to Reviewer XodB (4/4)**
>
> **(ii) Discussions with the robustness of backdoor attack, model fine-tuning, and continual learning methods**
>
> > **Weakness 3**:The paper investigates the vulnerabilities of model merging methods. The idea is similar to the robustness of backdoor attacks in model fine-tuning and continual learning. It would be great if the paper could compare these topics and discuss the major distinctions and unique challenges in model merging.
>
> We would like to clarify that our setting is indeed different from your mentioned robustness of backdoor attacks, model fine-tuning, and continual learning methods. **The core theme of our work is related to model merging using different models (partial backdoor) rather than one model like your mentioned works.**
>
> For the traditional backdoor attack, the model provided by the adversary is the final model for deployment. However, for model merging, the adversary only contributes to parts of the models, which are provided for the latter model merging, and the adversary has blind knowledge about how model merging is conducted. We are the first to explore the backdoor effect (backdoor succession and backdoor transfer) during model merging and provide a defense-aware merging method to mitigate this issue.
>
> **Model merging has its unique challenges compared with finetuning-based methods and continuous learning methods.**
>
> **Problem:** Fine-tuning-based methods directly add perturbations to the final model during fine-tuning[1-5], continual learning methods[6-9] additionally consider the forgetting issues related to backdoor attacks during sequential training, but both of them only focus on the optimization of its single model during training. In contrast, model merging should additionally consider the interference from other task-specific models. As shown in Figure 4, masking the backdoor-related parameters will also influence the parameters of task interference. There exists a trade-off between performance and safety due to the conflict of these two parts of parameters, which is special for model merging.
>
>  **Training Data Available**: As shown in Table 1,  we only have the unlabeled test data for model merging, but for finetuning-based and continual learning methods, they need some labeled data at least, which means the setting of model merging is different and difficult.
>
> **(iii) How does the Backdoor Detection Mask differ from existing trigger inversion or synthesis methods?**
>
> >**Question 1**: How does the Backdoor Detection Mask differ from existing trigger inversion or synthesis methods?
>
> Exactly, the object of the Backdoor Detection Mask is the same as existing trigger inversion or synthesis methods[10-11], aiming to find a backdoor trigger inserted into the model. **The key difference among them exists in the optimization process and special optimization constraints.**
>
> For example, some trigger inversion works need to recover the backdoor through an optimization process to flip a support set of clean images into the target class (e.g.smoothinv [10]) and other works (e.g. BEAGLE[11]) propose model backdoor forensics techniques and need a few attack samples as instructions. For our proposed DAM as shown in Table 1, the optimization of the backdoor detection mask only needs unlabeled test data. Simultaneously, as shown in Figure 4, both the Backdoor Detection Mask and the Task-Shared Mask contribute to the whole merging process and they are optimized alternately in an iterative process to develop a merged model that effectively balances performance and safety.
>
> [1] NIPS2021, Adversarial neuron pruning purifies backdoored deep models
>
> [2] NIPS2022, One-shot neural backdoor erasing via adversarial weight masking
>
> [3] NIPS2023, Shared adversarial unlearning: Backdoor mitigation by unlearning shared adversarial examples
>
> [4] ICCV2023, Enhancing fine-tuning based backdoor defense with sharpness-aware minimization
>
> [5] ICML2024, Vaccine: Perturbation-aware Alignment for Large Language Models against Harmful Fine-tuning Attack
>
> [6] ICML2023, Poisoning generative replay in continual learning to promote forgetting
>
> [7] CVPR2024, BrainWash: A Poisoning Attack to Forget in Continual Learning
>
> [8] Arxiv2024, Persistent Backdoor Attacks in Continual Learning
>
> [9] Arxiv2024, Adversarial Robust Memory-Based Continual Learner
>
> [10] CVPR2023, Single Image Backdoor Inversion via Robust Smoothed Classifiers
>
> [11] NDSS2023, Unlearning Backdoor Attacks through Gradient-Based Model Pruning

---

> > ### Author Response · Authors · 2024-11-27
> > **Waiting for discussions with Reviewer XodB**
> >
> > Dear Reviewer XodB,
> >
> > We sincerely appreciate your valuable suggestions to help us polish our paper. We have carefully considered each point and addressed them one by one in our rebuttal. The updated PDF has been uploaded.
> >
> > As the Author-Review Discussion period draws to a close, we would like to ensure that we have thoroughly addressed all of your concerns and resolved any outstanding issues. We remain open to any further comments or suggestions you may have and are willing to provide additional clarifications if necessary. Of course, we sincerely hope you can reconsider the score if the mentioned problems have been solved.
> >
> > Best regards,
> >
> > The Authors

---

> > > ### Author Response · Authors · 2024-12-02
> > > **Kind Reminder to Reviewer XodB**
> > >
> > > Dear Reviewer XodB,
> > >
> > > Thank you for your time and valuable feedback on our work. **As the reviewer reply period is coming to the end**, we kindly encourage you to share any remaining questions or concerns at your earliest convenience, as we will not be able to respond after this period.
> > >
> > > Your insights are greatly appreciated and have significantly contributed to improving our paper. Of course, we sincerely hope you can reconsider the score if the mentioned problems have been solved.
> > >
> > > Best regards,
> > >
> > > The Authors

---

> > > > ### Comment · Reviewer_XodB · 2024-12-02
> > > >
> > > > Thank you for your replies and for the discussions in the updated Appendix. They have addressed most of my concerns, and I’ll raise my score.

---

> > > > > ### Author Response · Authors · 2024-12-02
> > > > >
> > > > > Thanks again for your insightful feedback!

---

### Official Review · Reviewer_o55R · 2024-11-05

**Soundness:** 4
**Presentation:** 4
**Contribution:** 3
**Rating:** 8
**Confidence:** 3

**Summary:**

This paper investigates the poisoning of machine learning models in a merged model framework using backdoor attacks (the poisoned model infects the resulting merged model, or infects other clean models). It then proposes a method called DAM (Defense-Aware Merging) which identifies poisoned models and performs a 2-level optimization on both the accuracy (maximize) and attack success rate (minimize).

Experiments are performed using the CLIP Vision Transformer for 6 image classification tasks. Extensive evaluation is done, compared against 3 types of baselines (individual finetuning [3 baselines], multi task merging [10 baselines], and post-defense methods [3 baselines]. Additional experiments are performed on specific framework settings (number of attacks, configuration of poisoned and clean models).

The DAM method is better at lowering the attack success rate than increasing / maintaining the accuracy. Overall, this is a useful paper on not only adding to the literature of vulnerability of multi-task machine learning models but also proposing a functional solution to defending against such an attack.

**Strengths:**

1. There is sufficient theoretical grounding of the proposed method including background (section 2) as well as providing proofs and mathematical explanations (section 3)
2. There is extensive empirical evaluation in multiple configurations and against multiple types of baselines which is very convincing
3. Additional experiments are included in the appendix which further demonstrate the DAM's robustness
4. The paper is well-written and code / data are provided for reproducibility

**Weaknesses:**

The paper seeks to optimize both accuracy and attack success rate and while the DAM is good at minimizing the ASR, it is not so good in maintaining or increasing the accuracy. (Nevertheless, sufficient explanation and reasoning is given to explain the results and the task is quite challenging). So not a weakness per se, merely an observation.

**Questions:**

1. Why is the related work section in the appendix, and not part of the main paper?
2. Perhaps a sentence or two defining the accuracy and attack success rate will help clarify the evaluation metric

---

> ### Author Response · Authors · 2024-11-22
> **Reply to Reviewer o55R**
>
> We are grateful for your thoughtful feedback. Your assessment is valuable for us to polish our paper. We are glad to address your questions one by one.
>
> ***1、 The Position of Related Work***
>
> >**Question 1**: Why is the related work section in the appendix, and not part of the main paper?
>
> Thanks for your careful review. We indeed assign the part of related work in the Appendix, but this does not diminish the contributions of our work. Exactly, we have provided a detailed review of merging works related to our topic in the Preliminaries of Section 2.
>
> ***2、The Definition of Evaluation Metric***
>
> > **Question 2**: Perhaps a sentence or two defining the accuracy and attack success rate will help clarify the evaluation metric
>
> Thanks for your good suggestions. Take the image classification experiments on Vit as an  example, we add the definitions of ACC and ASR in our paper as follows:
>
> (i) ACC: The percentage of input images having no trigger classified into their corresponding correct classes(true label).
>
> (ii) ASR: The percentage of input images embedded with a trigger classified into the pre-defined target class(label decided by the attackers)
>
> For attackers, they aim to inject the backdoor into models (modifying parameters), resulting in backdoored models with high ACC and high ASR. In this way, they can confuse backdoored models with clean models for users(who mainly focus on basic ACC performance). Only when the specific triggers are added to the test image, the model will misclassify towards the pre-defined class. Thus, for defenders (like us), we aim to weaken the backdoor effect to acquire the model with high ACC but low ASR.
>
> We have added these clarifications referring to your valuable suggestions in the revised version.

---

### Author Response · Authors · 2024-11-26
**Looking Forward to Reply**

Dear All Reviewers,

Thank you for your invaluable feedback on our paper. Your comments and suggestions have been very helpful in improving the clarity and quality of our work.

As the discussion period deadline approaches, we would like to ensure that we have thoroughly addressed all of your concerns and resolved any outstanding issues. We remain open to any further comments or suggestions you may have and are more than willing to provide additional clarifications if necessary.

Best regards,

The Authors

---

### Author Response · Authors · 2024-12-01
**General Response to All Reviewrs**

We sincerely appreciate the reviewers for their thoughtful and constructive feedback. We are encouraged by the positive recognition of our contributions:

***(1) Technique Contributions***: The first work to reveal backdoor attack vulnerabilities in existing model merging methods, observing two interesting findings - backdoor succession and backdoor transfer and  proposing a reasonable meta-learning solution to address this problem (All Reviewers `o55R`, `XodB`, `3WWP`, `3WWP`) .

***(2) Extensive Experiments and Convincing Results***:  Extensive Experiments are conducted to clarify the meaning of the problem and the effectiveness of the proposed Defense-Aware Merging (All Reviewers, `o55R`, `XodB`, `3WWP`, `1tZq`).  We merge more than 6  task-specific models with 2 different backbones, with the comparisons including 3 types of baselines (individual finetuning [3 baselines], multi-task merging [10 baselines], and post-defense methods [3 baselines]. Moreover, additional experiments are performed on specific framework settings (number of attacks, configuration of poisoned and clean models) to verify the robustness and efficiency of the proposed method.

***(3) Writing***: The paper is well-written and easy to follow（Reviewers `o55R`, `3WWP`), with the sufficient theoretical grounding of the proposed method including background (section 2) as well as providing proofs and mathematical explanations (Section 3).


In our revision, we have carefully addressed each of the concerns raised:

***(1) More detailed statements for some definitions and settings*** (Reviewers `o55R`, `XodB`, `3WWP`):

- Provided detailed definitions of ACC/ASR and 'subspace" used in the paper.

- Provided clearer clarification for the backdoor succession and backdoor transfer: experimental results and theoretical statements from the perspective of disentangling the parameter components for the task vectors.

- Provided detailed attack and defense settings (the knowledge of the attackers and defenders) and a brief introduction to other backdoor-related work (e.g. BadVit, TrojVit, LoRA-as-an-Attack, WAG) to polish the paper.


***(2) Efficiency Analysis (The superiority of our proposed DAM against post-defense methods(ANP/AWM/SAU))*** (Reviewers `6SYF`, `XodB`):

- Provided detailed measurements of computational costs for post-defense methods(two-stage, merge first and then defense) and our proposed merge method( end2end training, defense during merging) to clarify the superiority of the proposed DAM compared with post-defense methods (ANP/AWM/SAU).

***(3) More detailed discussion about the Domain Source of models used for merging***  (Reviewers  `3WWP`):

- Provided detailed discussion for the domain source of the used models and conducted additional experiments to strengthen our contribution.

***(4) More detailed discussion about other related work(The robustness of backdoor attack, model fine-tuning, and continual learning methods*** (Reviewers   `XodB`):

- Provided detailed discussions about the major distinctions of model merging compared with previous works(The robustness of backdoor attack, model fine-tuning, and continual learning) from the optimized object(merging multiple models without tuning to achieve the target model rather than directly operating on the target model).

- Clarified unique challenges in model merging compared with previous works from the perspective of problem and training data available.
Simultaneously, disentangling the parameter components during merging can further verify the unique challenge(backdoor transfer) during merging.

We have also provided point-by-point responses to each reviewer's specific concerns. The updated paper has been uploaded (with the corrections in bold blue). We believe these revisions strengthen our paper while addressing all feedback. We welcome any further discussion from the reviewers.

---

### Meta-Review · Area_Chair_6jgd · 2024-12-23

**Metareview:**

This paper focuses on defending against backdoor attacks in merged ensembles of underlying base models.  The attack vector is that, e.g., I may have some models that I’ve trained from scratch on my own, but others that have been downloaded from Huggingface or another open source model repo that were trained/fine-tuned by an adversary to include a backdoor.  It’s known those backdoors can propagate after model merge, and the technique provided here gives an (imperfect) multi-step approach to blocking some of those backdoor attacks.  Reviewers appreciate the straightforwardness of the approach and the extensive experimental testing.  As with any paper in this space, there is concern that any defense mechanism without some form of certifiability may be incorporated into the next generation of adversary’s backdoor strategies.

**Additional Comments On Reviewer Discussion:**

Reviewers were responsive during the rebuttal phase, with clarifications asked about definitions in the paper as well as some additional back-and-forth on experimental tests/ablations.

---

### Decision · Program_Chairs · 2025-01-22

Accept (Poster)